biophysics

molecular motors, intracellular transport, collective behaviour, modelling

**Author for correspondence:**
Rhoda J. Hawkins
e-mail: rhoda.hawkins@physics.org

# Modelling cytoskeletal transport by clusters of non-processive molecular motors with limited binding sites

Naruemon Rueangkham, Ian D. Estabrook and Rhoda J. Hawkins

Department of Physics and Astronomy, University of Sheffield, Hicks Building, Hounsfield Road, Sheffield S3 7RH, UK

RJH, 0000-0002-9027-7622

Molecular motors are responsible for intracellular transport of a variety of biological cargo. We consider the collective behaviour of a finite number of motors attached on a cargo. We extend previous analytical work on processive motors to the case of non-processive motors, which stochastically bind on and off cytoskeletal filaments with a limited number of binding sites available. Physically, motors attached to a cargo cannot bind anywhere along the filaments, so the number of accessible binding sites on the filament should be limited. Thus, we analytically study the distribution and the velocity of a cluster of non-processive motors with limited number of binding sites. To validate our analytical results and to go beyond the level of detail possible analytically, we perform Monte Carlo latticed based stochastic simulations. In particular, in our simulations, we include sequence preservation of motors performing stepping and binding obeying a simple exclusion process. We find that limiting the number of binding sites reduces the probability of non-processive motors binding but has a relatively small effect on force–velocity relations. Our analytical and stochastic simulation results compare well to published data from *in vitro* and *in vivo* experiments.

# 1. Introduction

Molecular motors that facilitate intracellular transport can move cargoes such as vesicles, lipid droplets or mitochondria [1–7]. Motors bound to cargo can work either alone or cooperatively.

Transport by a single molecular motor has been widely studied both in experiments [8–12] and theory [2,4,13–17]. However, experiments *in vivo*, have revealed, by electron microscopy and by tracking of cargoes with optical methods, that in cells cargoes are generally transported by several motors [3,4,10,18–21]. Many studies have demonstrated that the number of molecular motors influences cargo transport, affecting the velocity, direction and persistence of cargoes [3,10,22–24].

In general, molecular motors can be classified according to their processivity, which refers to the distance that they can move along a filament before detaching [2,25]. Processive motors can individually interact with a filament for long times performing a large number of steps before detaching, i.e. they rarely unbind from the filament. Examples of processive motors include kinesin, which moves towards the plus-ends of microtubules (anterograde direction) and dynein, which moves towards the minus-ends of microtubules (retrograde direction) [25]. By contrast, non-processive motors will often unbind [2], remaining on a filament for only a short time and moving only a short distance. An example of a non-processive motor is myosin, which moves along actin filaments. In experimental biology, the distinction between processive and non-processive is often defined as between rare and common unbinding of motors respectively. However, in this paper, we use a purer definition of processive and non-processive motors to align with the qualitative difference between theoretical models. We assume 'processive' motors never unbind during the time frame under study whereas 'non-processive' motors are able to unbind and rebind to a filament within the time frame of the study. Feng *et al.* [26] show that reattachment kinetics play a dominant role in multimotor cargo transport by studying pairs of motors joined with DNA. While at first glance it may seem that processive motors are better, there may be considerable advantages of non-processive motors. For example, by considering hydrodynamics, Argentini & Lowe [27] suggest it could be energetically favourable for some fraction of motors to detach and be advected by fluid flow generated by bound motors.

Motors pulling on a cargo often pull against a force which affects their velocity. The force may be just the drag force caused by the cargo moving through the viscous medium or there may be additional external forces. For example, *in vitro* experiments can exert an external force on the cargo using optical tweezers or a magnetic field [10,28,29]. Various mathematical models have been introduced to explain the dynamics of collective motors pulling against a force from different perspectives [2,4,14,30–33]. In particular, two different models for the way force affects collective motors published in [2–4,14] have both been widely used, but confusion exists in the literature in applying these models. The former [2] introduced the leading motor model in which all the load force from the cargo is exerted on the leading motor which seems to be supported by some experiments [10,29]. By contrast, the latter [3,4,14], used a mean-field theory, assuming that all motors share the force equally, to analytically calculate the velocity of a cluster of motors moving along a cytoskeletal filament and seems supported by other experiments [34–36]. We discuss these two models in more detail in §2.1. The collective behaviour and cooperativity of multiple motors simultaneously pulling on a cargo is not yet fully understood and interesting research questions remain.

In this work, we model molecular motors moving along cytoskeleton filaments by using analytical expressions and Monte Carlo simulations. We compare our simulation results with mathematical models for a fixed number of motors on a single track. Campàs *et al.* [2] derived the analytical result for the velocity against force for a cluster of such processive motors. We extend this leading motor model to obtain an analytical expression for the case of non-processive motors using a similar method to that used in [4] for their mean field model. In addition, we mathematically and computationally study the case of a finite number of available binding sites on the filament and include the sequence preserving effects of simple exclusion in our simulations. Finally, we compare our analytical and simulation results to published experimental data [10,37].

# 2. Processive motors

## 2.1. Mathematical description of transport by collective processive motors

The standard model for multiple motor transport is based on the asymmetric simple exclusion process (ASEP) model [38–41]. The basic principle is to consider a filament as a one-dimensional lattice of sites that motors can occupy. Over time a motor can step forwards or backwards with specified rates but cannot move to an already occupied site. The stepping rates simplify the kinetics of the enzyme cycle producing a step into a single rate parameter. A more detailed study [32] of Michaelis–Menton kinetics of the enzyme cycle reveals a substrate dependence and enhanced velocity due to activation

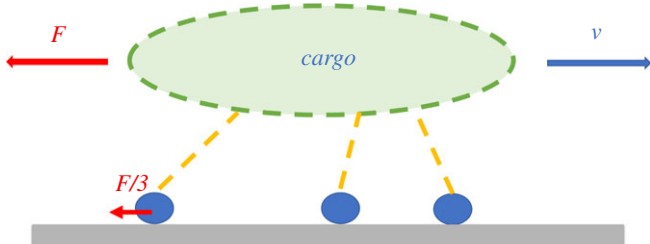

**Figure 1.** Sketch showing three motors (blue circles) attached to a cargo. The cargo moves with velocity *v* and the force *F* is shared equally by all three motors [14].

of motors before the next site is available. Using a single rate parameter as we do assumes the activation is immediate corresponding to high substrate/energy availability, i.e. we assume the ATP concentration is not limiting.

Over the past decade theoretical investigations of transport by multiple molecular motors under a load force have resulted in two widely used theoretical models for how force affects the motors, namely a mean-field theory [4] and a leading motor model [2]. However, which of these two models is more appropriate in particular applications remains ambiguous. Below we clarify the difference between these models and the assumptions made by them.

### 2.1.1. Mean-field model

Klumpp & Lipowsky [4] proposed a mathematical model for a cargo transported by $n$ molecular motors along a filament, assuming that the external load force $F$, is applied equally to the $n$ bound motors, such that an individual motor experiences a force $F/n$ (figure 1). Klumpp & Lipowsky [4] assume the velocity is a linear function of force and later Kunwar & Mogilner [14] extended the linear force–velocity relation to a more general nonlinear form with an exponent $w$, namely

$$V_n(F) = v\left(1 - \left(\frac{F}{F_s n}\right)^w\right),\tag{2.1}$$

where $v$ is the velocity when no load is applied and $F_s$ is the stall force of a single motor (the force for which the cargo stops) [1,14,29,42–45].

This theoretical model is suitable to apply to experiments in which the external force is exerted in such a way that all the motors share it equally. However, in many scenarios, such as in a cell, motor transport is resisted by an internal viscous drag force on the cargo from a fluid medium such as the cytoplasm. Such a force may be all experienced by the leading motor with motors following behind force free. Optical trap force measurements on kinesin-1 by Furuta *et al.* [10] using a polystyrene bead stuck to the motor assembly imply that one kinesin bears all the load. In the next section, we therefore review the leading motor model developed by Campàs *et al.* [2] which is more appropriate for such scenarios.

### 2.1.2. The leading motor model

Campàs *et al.* [2] developed an asymmetric simple exclusion process (ASEP) [38–41] for cargo dynamics driven by processive molecular motors. As illustrated in figure 2, each motor is able to move stochastically with specified forward, $p$, and backward, $q$, rates, similar to classical Brownian ratchet models [46–48]. The hopping rates for the leading motor are dependent on the force, [49], by a Boltzmann weighting; $p_1 = pe^{-f\delta}$ and $q_1 = qe^{f(1-\delta)}$, where $f$ is the dimensionless force exerted by the cargo such that the load force $F = f k_B T/\mathrm{d}x$ where $\mathrm{d}x$ is the motor step size. The dimensionless parameter, $\delta$, varies from 0 to 1 and determines how much the forward versus backward stepping rates are affected by the load force. The forward and backward rates of all the motors following the leading motor are equal and independent of force, i.e. $p_\mu = p$ and $v_\mu = v$ for $\mu > 1$ [2,40]. The analytical expression derived by Campàs *et al.* [2] for the average velocity of a cluster of $n$ motors is given by

$$V_n = p\frac{(1 - e^f(q/p)^n)(1 - q/p)}{e^{f\delta}(1 - q/p) + e^f(q/p - (q/p)^n)}.\tag{2.2}$$

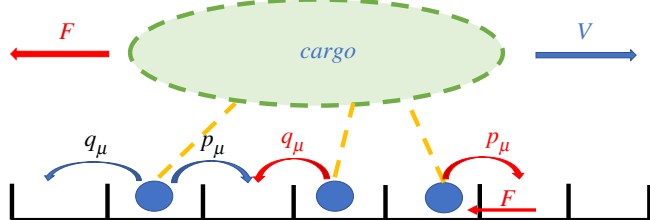

**Figure 2.** Possible motor transitions and associated rates. Motor tails are attached to a cargo whereas motor heads (blue circles) are attached to a filament. The boxes represent a one-dimensional lattice of binding sites on the filament [2].

In the simple case of unidirectional motors for which $q = 0$, equation (2.2) reduces to $V_n = pe^{-f\delta}$. For small forces, this can be expanded to reproduce the linear version of the mean-field theory in equation (2.1) with $\omega = 1$. We also note that a rough approximation of equation (2.2) can be obtained to describe the behaviour of multiple motors with a simpler model of a single motor with modified backwards stepping rates (see electronic supplementary material, appendix E). However, the validity of this approximation does not hold well for the parameter regions we are interested in.

Campàs *et al.* [2] also consider attractive and repulsive interactions between motors. Despite a small amount of evidence suggesting very weak attractive interactions [50], it is not yet clear what interactions are present experimentally. We therefore focus on the neutral case of no interactions between motors for our analysis.

As noted above, the leading motor model is suitable for situations such as a viscous drag force of a cargo *in vivo*, for which the total force is experienced by the leading motor. In addition, this model can easily be extended to other forms of unequal loading and as such is more general than the mean-field model, with the latter being a limiting case of the leading motor model. Kunwar *et al.* [1] present a more detailed model of motors as springs which results in motors sharing the force equally when at the same position and the leading motor experiencing most of the force when it is ahead. In our model, we consider simple exclusion and therefore do not allow multiple motors at the same site, so the leading motor is the most appropriate model for us to use. Therefore, in this work, we use this leading motor model to compare to *in vivo* experimental data [37].

We first simulate fully processive motors with the assumption that the leading motor experiences all the force. Then in §3, we extend this model to include motor attachment and detachment to study the case of non-processive motors. Campàs *et al.* [51] studied this extension to non-processive motors using simulations but did not do the analytical extension which we do in §3.

## 2.2. Simulations of processive motors

We first simulate the model of $N$ processive motors which do not unbind from the filament. We model the filament as an one-dimensional lattice with lattice spacing equal to the motor step size, $dx$. We initialize the simulation by randomly placing $N$ motors within the width of cargo. We choose the initial cargo width to be 10 lattice sites for $N < 10$ and $(N + 10)$ lattice sites for $N \geq 10$. To check this choice of initial conditions does not influence our results, we varied the width of the cargo between 10 and 100 and found that our results are insensitive to the chosen initial cargo width.

The simple exclusion rule dictates that each site may be either occupied or empty but cannot be occupied by more than one motor. Motors can thus move to a neighbouring site only if the new site is unoccupied. In the simulation, an individual motor moves forward with a rate $p_\mu$ and backward with a rate, $q_\mu$, where the subscript represents the motor $\mu$ (figure 2).

For consistency throughout this manuscript, we present results from our simulations using a discrete fixed time-step Monte Carlo method. However, since continuous-time discrete-state Markov processes such as the ones we study here are often stochastically simulated with the well-known event-driven approach of Gillespie [52], we checked our method against this. We found that our results using the fixed time-step and Gillespie algorithm methods are equivalent within the error bars (see results presented in electronic supplementary material, appendix A). We use our fixed time-step method for our full study since it is 10 times faster than the Gillespie algorithm for our non-processive motors case. In our fixed time-step method, at each time-step, $dt$, forward and backward steps are attempted with the probabilities $P_f = p_\mu \, dt$ and $P_b = q_\mu \, dt$, respectively. If motor $\mu$ meets the condition for moving to a neighbouring site, the position of the motor $x_\mu$ is updated to $x_\mu \pm dx$ for motion forwards (+) or backwards (−) along the track, where $dx$ is the motor step size.

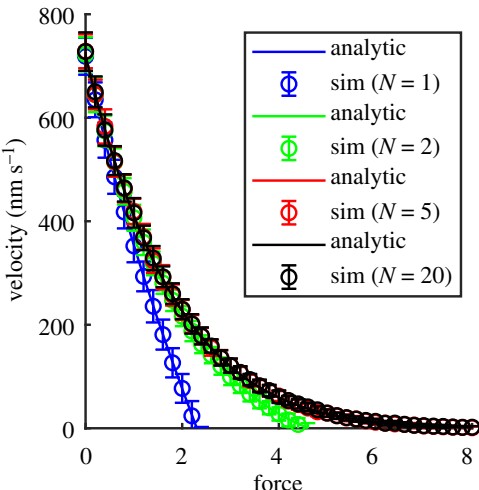

**Figure 3.** Force–velocity curves of processive motors from Monte Carlo simulations (symbols) for different number of motors $N = 1$ (blue), 2 (green), 5 (red) and 20 (black) compared with the analytical solution (lines) given by equation (2.2) with parameters $p = 100$ s$^{-1}$, $q = 10$ s$^{-1}$, $\delta = 0.5$ [2,9,10]. The force on the $x$-axis is the dimensionless force $f = F\mathrm{d}x/k_BT$, where $F$ is the physical force and $\mathrm{d}x = 8$ nm is the step size.

## 2.3. Results for processive motors

Here, we present our results for fully processive motors, which do not unbind from the track and may move bidirectionally with the rates $p$ and $q$. The case of unidirectional motors is easily obtained by setting $q = 0$. Our models can be applied to a variety of different molecular motors. The simulation results we show here are for parameter values $p = 100$ s$^{-1}$ and $q = 10$ s$^{-1}$ corresponding to experimental values for processive kinesin-1 [9,10] and the assumption that the ratio of forward to backward rates is 10 [53,54].

To compare with the analytical solution, equation (2.2), we calculate the steady-state velocity of the leading motor from our simulation results, $V_{\mathrm{sim}} = (x_1(T) - x_1(t_s))/(T - t_s)$, where $x_1(t)$ is the position of the leading motor at time $t$, $T$ is the total simulation time and $t_s$ is the time we start measuring from once the system has reached steady state. We find that running our simulations for $T = 5 \times 10^6$ time-steps of $\mathrm{d}t = 1.0 \times 10^{-3}$ s is long enough to ensure that the results are not dependent on specific initial conditions chosen. We choose $t_s = 1000$ steps by which time all cases have reached steady state (checked by plotting trajectories, not shown). We ran each simulation 100 times and plot the standard deviation as error bars in the results graphs.

The leading motor's velocity for different numbers, $N$, of bound motors is shown in figure 3 using our fixed time-step Monte Carlo method. It is clear that the velocities for different number of motors $N$ decrease rapidly with force as also seen in [2,33]. The force at which the velocity reaches zero (hits the $x$-axis in figure 3) is known as the stall force. For large $N$ and dimensionless force, $f$, the motors form a dense cluster which moves very slowly. Furthermore, these results imply that the force–velocity curves for multiple motors ($N > 5$) are almost indistinguishable.

To clarify the effect of number of motors $N$ on the velocity of the collection of motors, we plot the velocity against the number of processive motors under different loads in figure 4. The figure shows the leading motor's velocity versus different total numbers of motors $N$ from 1 to 10 for different loads (various values of dimensionless force $f = 0$, 2 and 4). The results in this figure confirm that the velocities of collections of $N > 1$ motors pulling the same force, $f$, are almost independent of the exact number of motors $N$. However, for $N \sim 1$ and finite force, the velocity is dependent on the number of motors and for large forces the velocity may be backwards (negative).

# 3. Non-processive motors

We now extend the theory and simulations described above to the case of non-processive motors. We allow motors to bind on and off the filament with rates $k_{\mathrm{on}}$ and $k_{\mathrm{off}}$, respectively, as illustrated in figure 5. Including the dynamics of motor binding and unbinding means that, unlike the processive motor case, the number of motors bound to a filament changes over time.

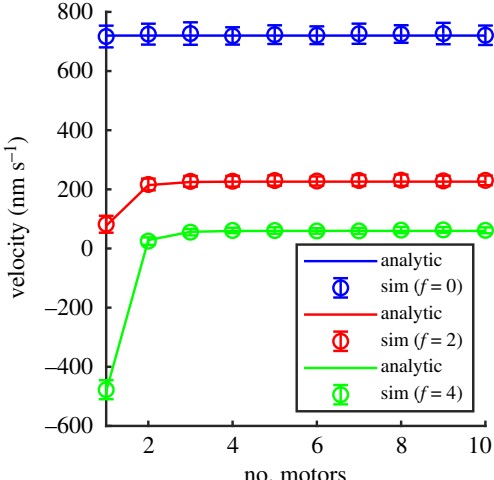

**Figure 4.** Velocity against number of processive motors under different loads (blue $f = 0$, red $f = 2$ and green $f = 4$) from our fixed time-step Monte Carlo simulations (symbols) compared with the analytical solution (lines) given by equation (2.2) with $p = 100$ s$^{-1}$, $q = 10$ s$^{-1}$, $\delta = 0.5$ and d$x = 8$ nm [2,9,10].

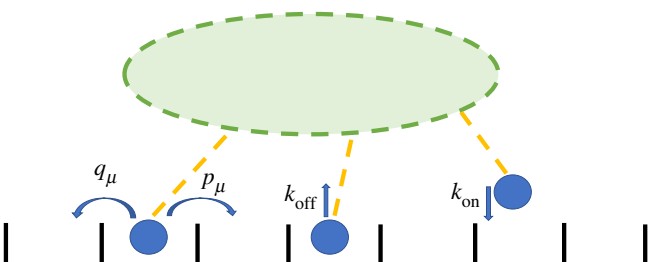

**Figure 5.** Cartoon showing motors bind to and unbind from a filament with rates $k_{\mathrm{on}}$ and $k_{\mathrm{off}}$, respectively.

## 3.1. Mathematical description of collective transport by non-processive motors

Since including stochastic binding and unbinding of motors (with rates $k_{\mathrm{on}}$ and $k_{\mathrm{off}}$) means the number of bound motors changes over time, we need to know the probability distribution of the number of bound motors. The average velocity of a cluster of motors capable of binding and unbinding can be calculated in terms of the probability, $P_n$ of having $n$ motors bound [4];

$$\bar{V} = \sum_{n=1}^{N} \frac{P_n V_n}{1 - P_0},$$ (3.1)

where $V_n$ is the velocity when there are $n$ motors bound given by equation (2.1) or (2.2) and $N$ is the total number of motors in the system. In using this equation, we assume that each cluster of $n$ motors travels with its steady-state velocity $V_n$. This assumption is valid as long as stepping is fast enough compared to (un)binding. In our work, we use equation (3.1), to calculate the average velocity of a cluster of non-processive motors and we calculate the probability distribution, $P_n$, as detailed in the following.

Initially, we assume that the number of binding sites available is unlimited such that the binding rate of a single motor, $k_{\mathrm{on}}$, is constant. We write down a discrete master equation for the probability, $P_n$, that there are $n$ motors bound at time $t$;

$$\frac{\partial P_n}{\partial t} = (N - n + 1)k_{\mathrm{on}}P_{n-1} + (n + 1)k_{\mathrm{off}}P_{n+1}$$
$$- (N - n)k_{\mathrm{on}}P_n - nk_{\mathrm{off}}P_n,$$ (3.2)

where $N$ is the total number of motors in the system (bound plus unbound motors). This master equation is equivalent to that used by Klumpp & Lipowsky [4].

At steady state, $\partial P_n / \partial t = 0$, we obtain the detailed balance condition that the binding and unbinding processes occur with equal probability such that

$$(N - n + 1)k_{\text{on}}P_{n-1} = nk_{\text{off}}P_n. \tag{3.3}$$

The probability, $P_n$, of being in a state with $n$ bound motors can therefore be written in terms of the binding rates $k_{\text{on}}$ and $k_{\text{off}}$ and the total number of motors, $N$. Using equation (3.3) and the normalization $\sum_{n=0}^{N} P_n(N) = 1$, we obtain

$$P_n(N) = \frac{N!}{n!(N - n)!} \left(\frac{k_{\text{on}}}{k_{\text{off}}}\right)^n P_0, \tag{3.4}$$

where $P_0$ is the normalization constant. If $k_{\text{on}} < k_{\text{off}}$ this can be written as a convergent series, $P_0 = (1 + (k_{\text{on}}/k_{\text{off}}))^{-N}$, and the maximum number of bound motors, $N$, can be infinitely large. The opposite case of $k_{\text{on}} > k_{\text{off}}$ will become incompatible with the initial assumption of constant $k_{\text{on}}$ as this will break down as the filament becomes saturated with motors. For that case, the finite number of binding sites available needs to be addressed by limiting the binding sites, as addressed in the next paragraph. The distribution (3.4) is the binomial distribution used in [55,56] and understood from considering the stochastic binding and unbinding of each motor when the maximum number of motors that can bind is $N$.

Considering the case of multiple motors attached to the same cargo we expect the number of filament binding sites accessible to those motors to be limited by the size of the motors and cargo. We therefore study the effects of limiting the number of binding sites. However, we do this without going into the level of detail of modelling motors as springs as some authors have done [1,14]. Instead we simply limit the binding sites accessible to the motors. If the number of binding sites on the filament is limited to $M$, which can be considered as the width of cargo, then $k_{\text{on}}(M, n) = (M - n)k_{\text{on}}^s$ where the superscript $s$ refers to the binding rate per site, obtained by dividing the unlimited binding rate by $M$, i.e. $k_{\text{on}}^s = k_{\text{on}}/M$. Equation (3.2) can be extended by including the limited number of binding sites, $M$ as follows:

$$\frac{\partial P_n}{\partial t} = (N - n + 1)(M - n + 1)k_{\text{on}}^s P_{n-1} + (n + 1)k_{\text{off}}P_{n+1}$$
$$- (N - n)(M - n)k_{\text{on}}^s P_n - nk_{\text{off}}P_n. \tag{3.5}$$

The distribution of bound motors in this case is given by

$$P_n(N, M) = \frac{N!M!}{n!(N - n)!(M - n)!} \left(\frac{k_{\text{on}}^s}{k_{\text{off}}}\right)^n P_0, \tag{3.6}$$

where $P_0$ is the normalization constant given by $\sum_{n=0}^{N} P_n(N, M) = 1$.

The average velocity of $N$ non-processive motors is then obtained by substituting equations (2.2) and (3.4) into equation (3.1) for unlimited binding sites giving

$$\bar{V} = \sum_{n=1}^{N} \frac{N!}{n!(N - n)!} \frac{P_0}{1 - P_0} \left(\frac{k_{\text{on}}}{k_{\text{off}}}\right)^n \left( p \frac{(1 - e^f(q/p)^n)(1 - (q/p))}{e^{f\delta}(1 - (q/p)) + e^f(q/p - (q/p)^n)} \right). \tag{3.7}$$

For the case of limited binding sites, the equivalent expression is obtained by substituting equations (2.2) and (3.6) into equation (3.1).

## 3.2. Simulations of non-processive motors

We extend the fixed time-step Monte Carlo algorithm we used for processive motors (described in §2.2) to the system of $N$ non-processive motors (drawn in figure 5) and validate the results and efficiency against a Gillespie algorithm as shown in electronic supplementary material, appendix B.

We consider $N$ motors attached to a cargo, thus the number, $n$, of motors bound on the filament at time $t$ varies between zero and $N$. We choose an initial condition with all the motors unbound from the filament. Then, at subsequent time-steps, each motor is allowed to attach to the filament track with the attachment probability $P_{\text{on}} = k_{\text{on}} \, dt$.

In each time-step of the simulation, we visit each of the $N$ motors to consider their possible states and positions. Each motor is allowed to either bind to and unbind from the track according to the relevant probability, $P_{\text{on}} = k_{\text{on}} \, dt$ or $P_{\text{off}} = k_{\text{off}} \, dt$, respectively. After that, the bound motors are allowed to move along the track with the forward/backwards stepping probabilities and the constraints of the simple exclusion process in the same way as for the simulation of processive motors described in §2.2.

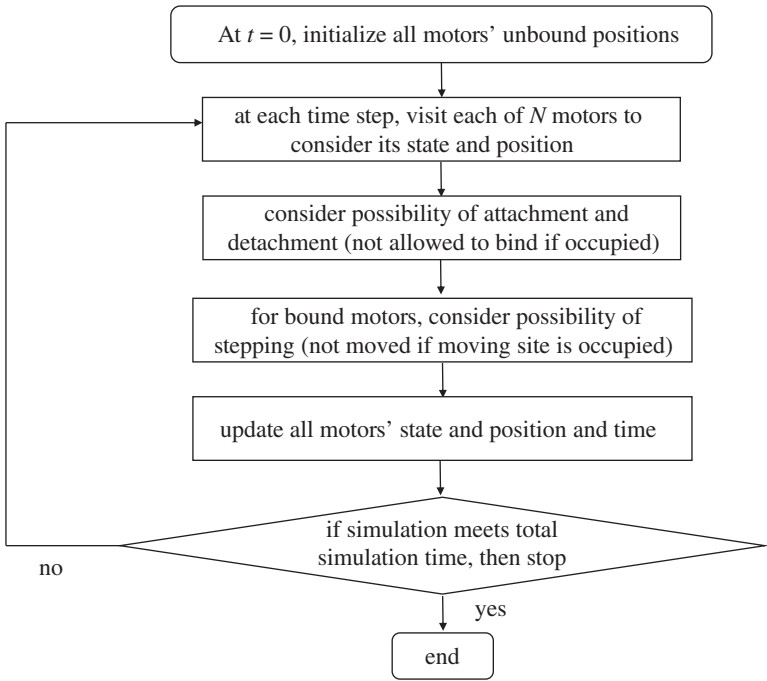

**Figure 6.** Flow chart of our fixed time-step Monte Carlo simulations for non-processive motors.

Therefore, the motors cannot overtake each other when stepping along the track. We summarize our procedure for simulating non-processive motors in a schematic in figure 6.

## 3.3. Results for non-processive motors

We now present the results of our fixed time-step Monte Carlo simulations for the stepping of a cluster of $N$ motors including the ability of binding on and off the filaments ($k_{on}$ and $k_{off}$). Since we are interested in motor clusters we study strongly binding molecular motors for which the binding rate, $k_{on}$, is higher than the unbinding rate, $k_{off}$ [4,10,57,58]. We choose Ncd (kinesin-14) as an example of a non-processive motor which unbinds often [10,59]. In our simulation, we use $k_{off} = 10$ s$^{-1}$ as measured experimentally [10,59,60]. $k_{on}$ is more difficult to measure experimentally, but, since we are interested in the case $k_{on} > k_{off}$, we choose $k_{on} = 20$ s$^{-1}$. We run our simulations for $5 \times 10^7$ time-steps with a time-step of $1.0 \times 10^{-3}$ s. This is longer than required for processive motors because non-processive motors take longer to reach steady state due to the binding dynamics. For non-processive motors, we set $t_s = 2500$ steps and $t_s = 25\,000$ steps by which time all cases have reached steady state for $N \leq 10$ and $N = 100$, respectively. In the following, we first present the distribution of the number of bound motors and then the average velocity, stall force and run length.

### 3.3.1. Probability distribution of bound motors

We consider the effect of number of binding sites in two scenarios. First, the number of binding sites is fixed in scenario A. Second, the number of binding sites is variable in scenario B and the sequence of motors is preserved on binding.

*Scenario A: Fixed number of binding sites.* We first assume that the number of binding sites accessible by the motors is fixed to $M$ lattice sites as illustrated in figure 7a. In this scenario, the motors are only able to bind within these $M$ sites, which can be considered to correspond to the cargo width. In this scenario, we allow an unbound motor to bind onto any of the unoccupied sites within the $M$ sites. This means that the sequence of motors may change during rebinding. In other words, in the simulation of this scenario motors are allowed to swap positions on rebinding. In figure 7b, we plot the probability distribution of bound motors, $P_n(N, M)$ for $N = 10$ and various different number of binding sites $M$. We choose $N = 10$ here to highlight the differences between limited and unlimited binomial distributions which are smaller for $N \leq 10$. The dark blue solid line is the analytical distribution for unlimited binding sites (equation (3.4)) and the dashed lines are the analytical distributions for the number of binding sites limited to $M$ (equation (3.6)) where the different colours represent different values of $M$. The

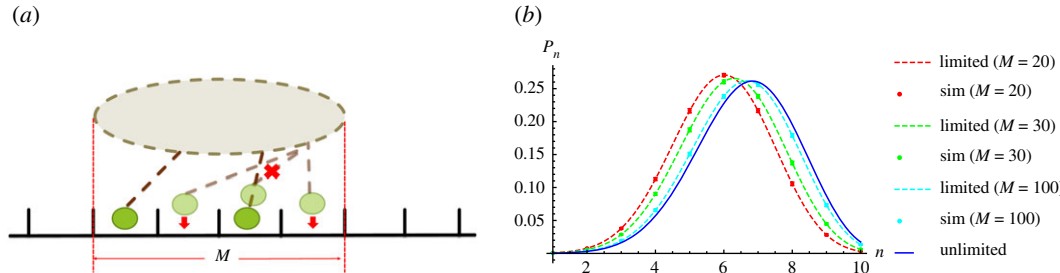

**Figure 7.** (a) Cartoon showing scenario A in which motors may bind only to unoccupied sites within a fixed limited number $M$ of binding sites but they may swap positions on rebinding. (b) Probability distribution of non-processive motors attached on the same cargo for various fixed number of binding sites ($M = 20$, 30, 100 for red, green, blue, respectively) from simulation results (points) and equation (3.4) (dark blue line) and equation (3.6) (dashed lines). The parameter values are $k_{on} = 20$ s$^{-1}$, $k_{off} = 10$ s$^{-1}$ [10,59,60], $p = 22$ s$^{-1}$ [10], $q = 2.2$ s$^{-1}$, $\delta = 0.5$, $f = 0$ and $N = 10$.

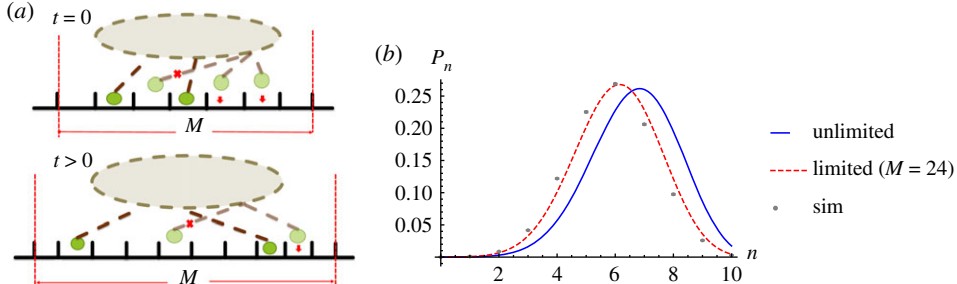

**Figure 8.** (a) Cartoon showing scenario B (variable number of binding sites and sequence preservation) in which motors may bind only to unoccupied sites between neighbouring motors such that the sequence is preserved. (b) Probability distribution of non-processive motors with no exerted force and variable number of binding sites, $M(t)$, from simulation results (points) comparing with equation (3.4) (blue line) and equation (3.6) (dashed line) with the value of $M$ being the average calculated over the whole simulation. The parameter values used are $k_{on} = 20$ s$^{-1}$, $k_{off} = 10$ s$^{-1}$ [10,59,60], $p = 22$ s$^{-1}$ [10], $q = 2.2$ s$^{-1}$, $\delta = 0.5$, $f = 0$ and $N = 10$.

coloured points are results from Monte Carlo simulations of scenario A (fixed $M$ binding sites) allowing the motors to swap positions on rebinding. The simulation results match the relevant analytical expressions (equation (3.6)) within the error bars. For $M \gg N$, the simulations and analytical distribution for the limited case (equation (3.6)) approach that of the unlimited case (equation (3.4)) as expected. Consequently, if the cargo size as defined by $M$ is large enough then we can describe the probability distribution of bound motors by the unlimited binomial distribution.

However, *in vivo*, due to steric hindrance, the molecular motors may preserve their sequence on rebinding. In this case, the number of accessible binding sites would change over time according to the positions of the leading and last motors. We consider this scenario in the following.

*Scenario B: Variable number of binding sites with sequence preservation.* Here, we present a second, more realistic, scenario in which the motor sequence is preserved on rebinding and the number of binding sites $M$ is no longer fixed. To preserve the sequence, a motor can only bind to unoccupied sites between its neighbouring motors. The number of accessible binding sites changes over time following the first and last motors' positions. Additionally, we allow one more site in front/behind the position of the leading/back motors to be accessible to stepping and binding. This latter accommodation allows for a motor to stretch to reach the adjacent site. At the end of each time-step, after motors have had the chance to bind and move, the number of binding sites $M$ is changed following the bound/moved leading and last motor positions. In the case that all motors are detached, $M$ is determined from the positions of leading/back motors when they were last attached. The number of binding sites $M(t)$ in this scenario is thus updated each time-step. In the case we consider, without interactions between motors, $M$ is limited to the steady-state cluster size plus 2.

The simulation results for $P(n)$ shown in figure 8 fit better with limited binomial distribution (equation (3.6)) than unlimited binomial distribution (equation (3.4)). However, the simulation results are shifted slightly lower than the analytical distribution because of the sequence preservation on motor rebinding in the simulation which is not included in the analytical expression. Electronic supplementary material,

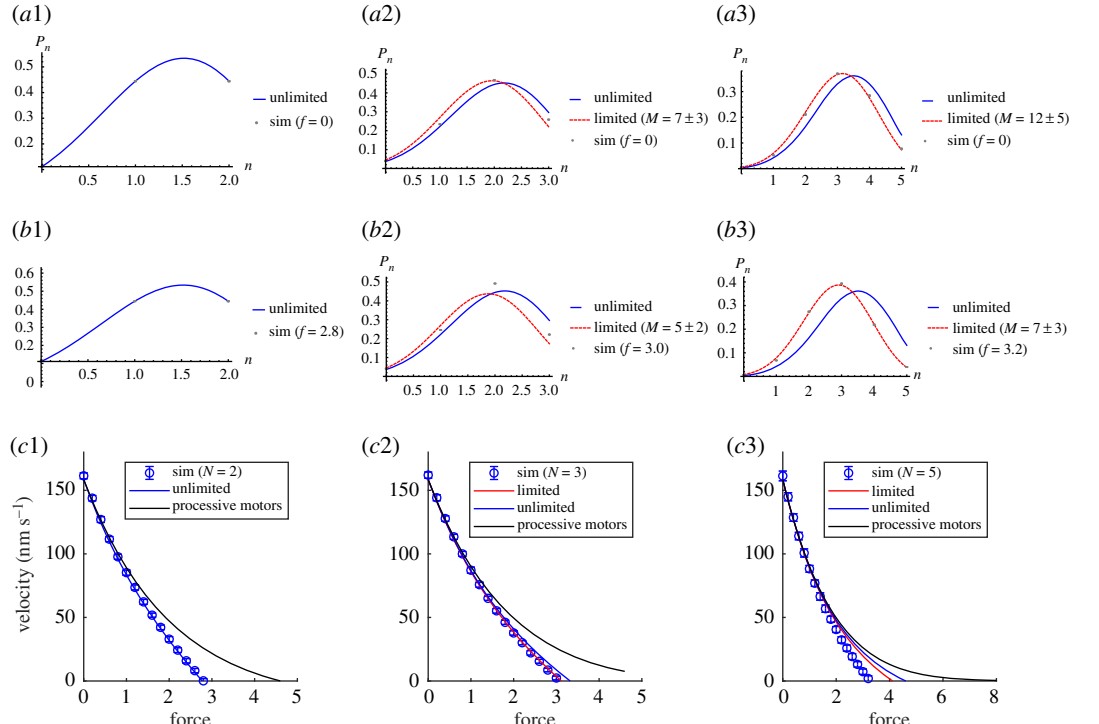

**Figure 9.** ($a1$–$b3$) Probability distribution of $n$ bound motors with maximum number of motors $N = 2$, 3 and 5 under ($a1$–$a3$) no force, $f = 0$ and ($b1$–$b3$) stall force, $f = f_s$, from simulations (symbols) compared with $P(n)$ given by equation (3.6) by averaging $M$ sites (dashed red) and equation (3.4) (solid blue). ($c1$–$c3$) Velocity of clusters of $N = 2$, 3 and 5 from simulation results (symbols) compared with the analytical solution equation (3.7) for limited (red) and unlimited (blue) binding sites. Note that, for $N = 2$, there are always available sites to bind, so it is not compared with limited binomial distribution. The solid black lines are the velocities of having $N = 2$, 3 and 5 bound processive motors without (un)binding, given by equation (2.2). The other parameter values are those for Ncd, i.e. the same as in figure 8, namely $k_{on} = 20$ s$^{-1}$, $k_{off} = 10$ s$^{-1}$, $p = 22$ s$^{-1}$, $q = 2.2$ s$^{-1}$, $\delta = 0.5$ and d$x = 8$ nm. For the processive case (black lines) $k_{on} = k_{off} = 0$.

figure S18 in appendix C shows that the results of this variable number of binding sites case without sequence preservation on rebinding do coincide with the analytical limited binomial distribution.

To date, there is no clear experimental evidence showing whether motors swap position or preserve their sequence during rebinding. However, most literature models use the assumption that the sequence of motors is preserved on binding [2,4,44,61]. Also, many of these studies do not support the assumption that motors are restricted to bind within a fixed determined space as is the case in our scenario A. We suggest that our scenario B (variable number of binding sites with sequence preservation) is more realistic for intracellular transport. We therefore choose scenario B to study further the velocity of a cluster of non-processive motors in the following section.

### 3.3.2. Average velocity

We compute the velocity of the motor cluster from the simulation by averaging the leading motor's velocity when at least one motor is bound on the track. The average velocity is then calculated by $V_{sim} = \langle (x(t_2) - x(t_1))/(t_2 - t_1) \rangle$ where $x(t_1)$ is the leading motor's position when it becomes the leading motor (due to it binding or the previous leading motor unbinding) and $x(t_2)$ is the leading motor's position when it stops being the leading motor (as it unbinds from the track or another motor binds in front of it). During the course of the simulation which motor is the leading motor will change according to the binding and unbinding of motors. The leading motor is the bound motor with the forwardmost position at that point in time, i.e. the largest $x_\mu(t)$. We calculate the velocity over the time that an individual motor acts as the leading motor ($t = t_1$ to $t_2$) and then calculate the velocity of the next motor that acts as the leading motor. We then average over all these velocities.

In order to obtain the force–velocity relation for non-processive motors, we first consider the appropriate steady-state probability distribution for the number of bound motors $P(n)$ at different forces and then consider the velocity at different forces. Figure 9 shows $P(n)$ from simulations and

analytical expressions (equations (3.4) and (3.6)) for (a) zero force, $f = 0$, and (b) the stall force, $f = f_s$ (the force for which the velocity is zero) for small numbers of total motors $N = 2$, 3 and 5. We choose values $k_{on} = 20 \text{ s}^{-1}$, $k_{off} = 10 \text{ s}^{-1}$, such that $k_{on} > k_{off}$ to study motor clusters. As discussed in the previous section, the analytical expression for the distribution of bound motors with limited number of binding sites, equation (3.6), is a better approximation to the simulation results than the unlimited binomial distribution, particularly at larger force. This is because the leading motor is more likely to move backwards at larger force, further reducing the number of accessible binding sites. Consequently, the probability distribution of motors bound on the track is shifted to smaller numbers of motors than the case assuming unlimited binding sites (equation (3.4)), as is evident in figure 9b2 and b3. The case of $N = 2$, shown in figure 9a1 and b1, is well described by the unlimited binding sites expression (equation (3.4)). This is because our model allows binding to one site in front/behind the leading/ back motor which means for $N = 2$ there are always available sites to bind. For $N = 3$, figure 9a2, the simulation points are between that of the limited and unlimited curves due to the extra freedom provided by the end sites. However, by $N = 5$, figure 9a3 we see a good agreement with the limited analytical expression. Note for $N = 10$ in figure 8b, the simulation points are shifted to smaller $n$ than the limited analytical expression. This is caused by the restrictions imposed in the simulations to ensure sequence preservation, which is not captured by the analytical expressions.

Figure 9c1–c3 shows the velocity as a function of force calculated from simulations and analytical calculations (equation (3.1)) for small numbers of total non-processive motors $N = 2$, 3 and 5. The processive motor case, equation (2.2), is plotted on the same graph for comparison. The parameter values used are those for Ncd for the non-processive case and $k_{on} = k_{off} = 0$ for the processive case (black line). See figure 9 caption for the full list of parameters. As expected from our studies of the probability distributions for the number of bound motors (figure 9a,b), the velocity for the case of limited number of binding sites is a closer approximation than that of unlimited binding sites. However, the effect on the velocity of limiting the number of binding sites is slight for small number of motors up to $N = 3$ (figure 9c2) with both analytical cases falling within the error bars of the simulations for almost the entire range of forces $f = 0$ to $f = f_s$. However, clear discrepancies develop for the $N = 5$ case (figure 9c3) at larger forces. Although the limited number of binding sites case is closer to the simulation results, even this is outside of the error bars for $N = 5$ at larger forces. This may be partly due to the larger forces decreasing the stepping rate such that the validity of steady-state assumption of equation (3.1) is weakened, resulting in the analytical solution overestimating the velocity. The discrepancy seen in figure 9c3 also reflects the influence of sequence preservation that is included in the simulations but not in the analytical results. This effect is more pronounced for larger clusters of motors and larger forces (see electronic supplementary material, figure S10 and appendix C).

Figure 10a1 and b1 shows the probability distribution of bound motors, $P(n)$, at no force, $f = 0$, and the stall force, $f = f_s$, for a larger motor cluster of $N = 10$. As expected, we see that the case of limited binding sites fits better than for unlimited binding sites for both extremes of the force range. In figure 10b1, we can see that the simulation points are shifted towards smaller number of bound motors compared to the analytical limited binding sites case. This is even more pronounced for the $N = 100$ case shown in figure 10a2 and b2. This is due to the effect of sequence preservation, which has a greater impact for larger clusters and larger forces. Electronic supplementary material, figure S18 in appendix C shows that without sequence preservation on rebinding the probability distributions of the number of bound motors follow that of the analytical expression for limited binding sites.

In figure 10c1, we show the velocity of a cluster of $N = 10$ non-processive motors with parameters for Ncd. We compare our simulation results against the analytical results for the cases of unlimited and limited binding sites as well as the case of 10 processive motors with $k_{on} = k_{off} = 0$. As can be seen in figure 10c1, all three analytical curves collapse onto that of the processive motor case for $N = 10$ indicating that for $N = 10$ there are enough motors in the cluster that they behave as processive motors. This processive behaviour of large clusters of non-processive motors can be justified analytically. Consider approximating the average velocity given by equation (3.1) by the velocity for the number of bound motors at the peak of the probability distribution for the number of bound motors. For unlimited number of binding sites, the peak of the relevant binomial distribution is at $n = Nk_{on}/k_{on} + k_{off}$. We see from figure 4 that for large number of motors the velocity is independent from the number of motors. Therefore, if $Nk_{on}/(k_{on} + k_{off}) \gg 1$ then $\bar{V} \approx V_{Nk_{on}/(k_{on}+k_{off})} \approx V_N$. Approximating the average velocity by that for the peak of the distribution for the number of bound motors is valid as long as the bulk of the probability distribution is for $n \gg 1$ bound motors since $V_n$ is independent of $n$ for $n \gg 1$. Figure 10a1 shows that this is already the case for $N = 10$. For $N = 100$, figure 10c2 shows that within the errors the velocity of $N = 100$ non-processive motors is as fast as that

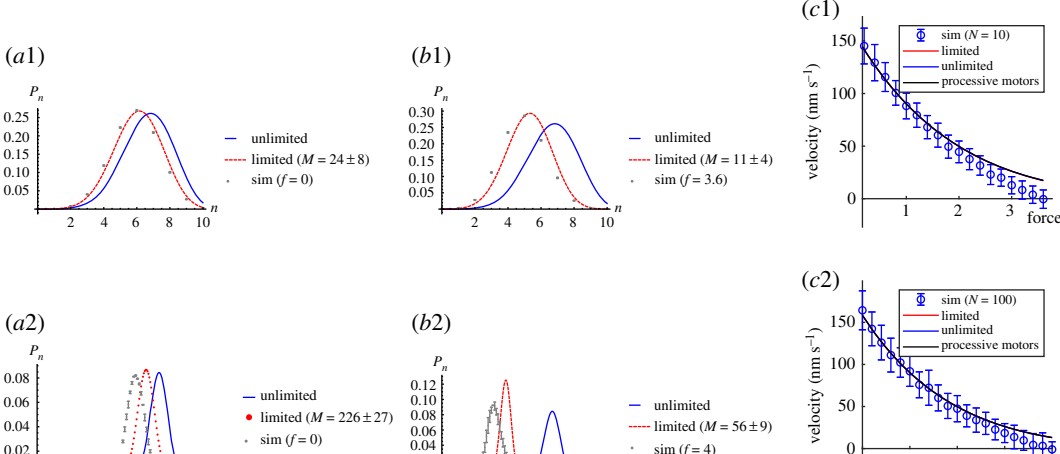

**Figure 10.** Probability distribution of $n$ bound motors with maximum number of motors $N = 10$ under ($a1$) no force, $f = 0$, and ($b1$) stall force, $f = f_s$; and $N = 100$ ($a2$) $f = 0$ and ($b2$) $f = f_s$, from simulation (symbols) compared with equation (3.6) by averaging $M$ sites (dashed red) and equation (3.4) (solid blue). ($c1$–$c2$) Velocity of large clusters of motors ($c1$) $N = 10$ and ($c2$) $N = 100$ are plotted and the analytical velocity of having $N$ bound processive motors without (un)binding (solid black) is also shown, given by equation (2.2). The other parameter values are the same as in figure 8, namely $k_{on} = 20$ s$^{-1}$, $k_{off} = 10$ s$^{-1}$, $p = 22$ s$^{-1}$, $q = 2.2$ s$^{-1}$, $\delta = 0.5$ and $dx = 8$ nm. For the processive case (black lines) $k_{on} = k_{off} = 0$.

of processive motors. This is because for $N = 100$ there is almost always at least one motor bound so, since the velocity of a cluster is limited by the leading motor, the effect of detachment is negligible. Therefore, we can approximate the velocity of a cluster of large enough numbers of non-processive motors as equivalent to the velocity of processive motors with the same number of motors.

In summary, our simulation results show that it is better to approximate the distribution of number of bound motors with equation (3.6) including the limited number of binding sites. This expression does not fit exactly because the simulation includes sequence preservation which is not included in the analytical theory. However, when we calculate the velocity of a cluster of $N < 10$ motors, both limited and unlimited analytical distributions can similarly approximate the velocity of a simulated motor cluster, see figure 9$c1$–$c3$. Therefore, when investigating the velocity, it is not crucial which distribution we use for small $N$. The distribution with unlimited number of binding sites has the advantage that it requires fewer parameters such that we only need to know the total number of motors $N$ and we do not need to know the size of the cargo nor number of binding sites $M$. This case of small $N < 10$ motor clusters bound to a single cargo has been shown to be relevant *in vivo* [18,62,63]. For large motor clusters $N > 10$, however, the distribution with unlimited number of binding sites is not a good approximation for the probability distribution of bound motors due to sequence preservation. However, since large clusters of $N > 10$ rarely have all motors unbound they behave like processive motors and the force–velocity curve can be described by that for processive motors.

For completeness, in figure 11$a$, we plot the velocity against the number of non-processive motors from simulations and analytical calculations using the unlimited binomial distribution. This is the non-processive motor equivalent of figure 4, which is for processive motors. Note that the values for the velocities are different for these cases due to the different forward and backward stepping rates used to mimic the molecular motors kinesin and Ncd. For comparison, we therefore also plot the case of processive motors (with $k_{on} = 0$ s$^{-1}$, $k_{off} = 0$ s$^{-1}$) and other parameters for Ncd on figure 11$a$ for simulation (cross symbols) and analytical expression, equation (2.2) (dashed lines). This comparison clearly supports the point that non-processive motors behave as processive motors for larger $N = 10$, consistent with figure 10$c1$. Figure 11$a$ shows the velocity increases with the maximum number of motors, $N$, but tends to a plateau for $N > 5$, suggesting that for $N > 5$ the binding of additional motors makes little difference to the velocity. This is similar to the case for processive motors (figure 4) except that for those the plateau is reached for $N > 1$. We also plot the distribution of these velocities for $N = 2$ with different forces in figure 11$b$. Discrepancies between the simulation and analytical results can be seen for the larger force $f = 4$, shown in light blue in figure 11$a$. This is due to sequence preservation included in the simulation but not in the analytical results. As expected, sequence preservation has a larger effect at larger forces. This can be seen by comparing figure 10$a1$ and $b1$

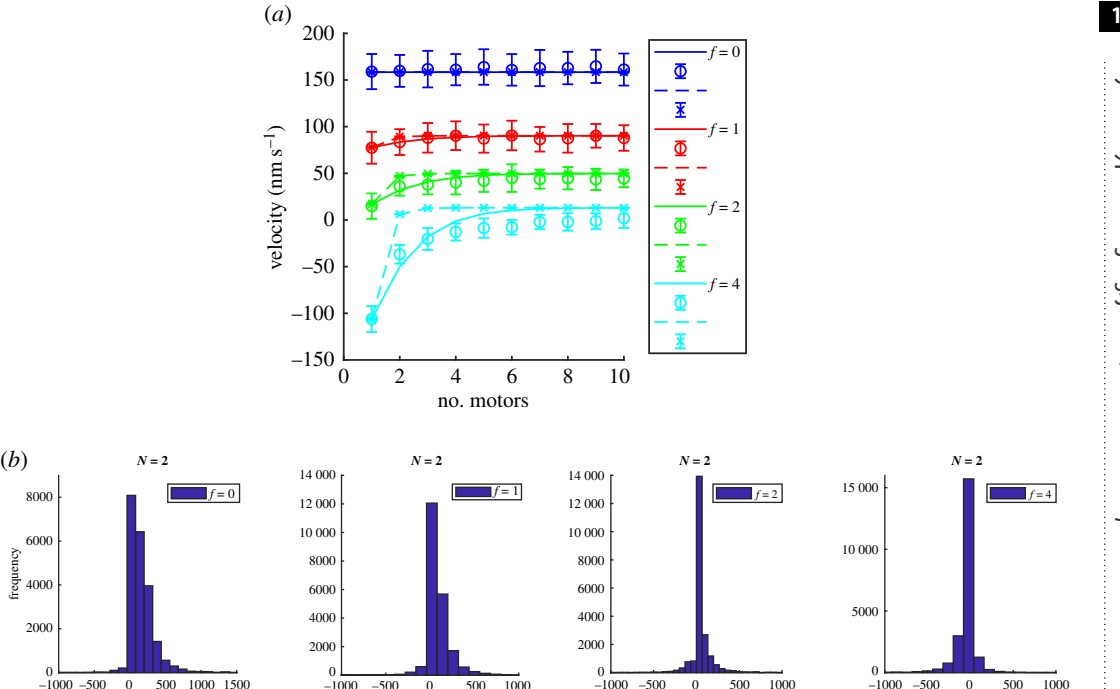

**Figure 11.** (a) Velocity against number of non-processive motors pulling a load with dimensionless force $f = 0$, 1, 2 and 4 (dark blue, red, green, light blue, respectively) from Monte Carlo simulations (circle symbols) and analytical expression using unlimited binomial distribution, equation (3.7) (solid lines). We also plot the case of processive motors (with $k_{on} = 0$ s$^{-1}$, $k_{off} = 0$ s$^{-1}$ simulation (cross symbols) and analytical expression, equation (2.2) (dashed lines). (b) The distribution of velocities in panel (a) for $N = 2$ with $f = 0$, 1, 2 and 4 (left to right). The other parameter values are the same as in figure 8, namely $k_{on} = 20$ s$^{-1}$, $k_{off} = 10$ s$^{-1}$, $p = 22$ s$^{-1}$, $q = 2.2$ s$^{-1}$, $\delta = 0.5$ and d$x = 8$ nm.

with electronic supplementary material, figure S18 in appendix C for $N = 10$ at $f = 0$ and $f = f_s$ with and without sequence preservation on rebinding.

### 3.3.3. Stall force against number of motors

The stall force is the force at which the cargo stops moving because the team of motors is no longer able to pull it against the load force. We can calculate the stall force, $F_s$, by substituting $V = 0$ into equation (2.2) for processive motors. For non-processive motors, we calculate the stall force from the force–velocity relations by setting equation (3.1) to zero using the unlimited binomial distribution and limited binomial distribution.

For processive motors, equation (2.2) can be solved analytically for the stall force giving $F_s(N) = NF_s(1)$ where $F_s(1)$ is the stall force of a single motor, i.e. for processive motors the stall force increases linearly with the number of motors. In dimensionless form, the stall force for a single motor is given by $f_s(1) = \ln(p/q)$ and is the same for both kinesin-1 and Ncd with the parameters we use for these different motors. We plot this in figure 12 along with the stall force for kinesin-1 extracted from our simulations displayed in figure 3. We use a threshold of $|V_{f_s}/V_{f_0}| \leq 10^{-5}$ to determine the stall force from our simulations.

For non-processive motors, $N$ is the maximum number of motors that can attach to the filament, so the number of attached motors at a particular time can vary from zero to $N$. Consequently, our numerical calculation of the stall force from the analytical result (equation (3.1)) for the stall force in the case of non-processive Ncd motors is lower than that of processive motors at the total number of motors $N$, as shown in figure 12. The numerical result for non-processive motors with unlimited number of binding sites increases linearly for large number of motors ($N > 10$) with the same gradient as that for processive motors (see electronic supplementary material, appendix D). This is also the case for binding sites limited to the average found in simulations. This is to be expected from our observations (§3.3.2) that clusters of $N$ non-processive motors behave like processive motors for large $N > 10$. In other words, once the number of motors is large enough, the stochasticity of (un)binding is effectively averaged out

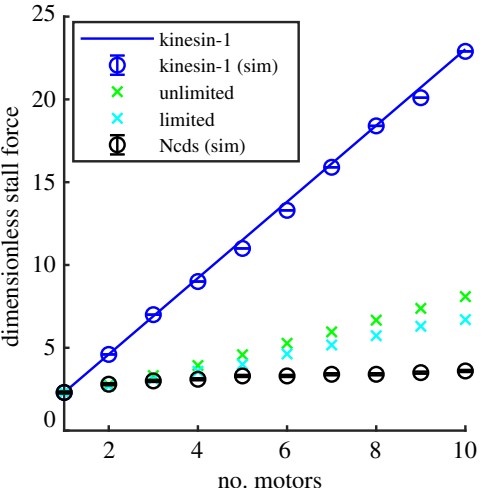

**Figure 12.** Dimensionless stall force against number of motors, $N$. For processive kinesin-1 motors, we calculate this analytically by setting equation (2.2) to zero (solid blue line). For non-processive Ncd motors, we set equation (3.1) to zero and solve numerically using the distribution of bound motors with limited (equation (3.6), light blue crosses) and unlimited (equation (3.4) green crosses) binding sites. Simulation results are shown with circle symbols for parameter values $p = 100\ s^{-1}$ and $q = 10\ s^{-1}$ for kinesin-1 processive motors (blue circles) and $p = 22\ s^{-1}$ $q = 2.2\ s^{-1}$, $k_{on} = 20\ s^{-1}$ and $k_{off} = 10\ s^{-1}$ for Ncd non-processive motors (black circles).

and no longer significantly affects the strength of the motor cluster. We therefore conclude that the larger the number of motors, $N$, bound to a cargo, the larger the load forces that can be overcome by the motors to transport a cargo for both processive and non-processive motors. Note that the stall force scales with the number of motors for cargoes such as beads [10] or filaments in motility assays [6,64], but that for fluid like cargo such as vesicles [23] or membrane tubes [51,60,65] it does not, since these cargo do not experience tangential forces [51].

Figure 12 shows that for small number of motors the stall forces for non-processive Ncds extracted from our simulations (black circles) correspond well with those numerically calculated from the analytical solutions with the number of binding sites unlimited (green crosses) and limited (light blue crosses). However, for $N > 3$, the results from our simulations are lower than those obtained from the analytical expressions and remain lower for $N > 10$, see electronic supplementary material, appendix D. This is due to the significant restrictions on binding due to the sequence preservation included in the simulations but not in the analytical results, as discussed in §3.3.2 and electronic supplementary material, appendix C.

### 3.3.4. Run length

We also examine the run length defined as the distance that the cargo moves until all the motors have detached from the filament. To find the average run length, we record the position, $x_1$, of the first motor to bind to the filament at time $t_1$ and the position, $x_2$, where the last remaining motor unbinds at time $t_2$. The difference between these positions gives the run length of the cargo. We take the average over all the periods when the cargo is bound to the filament. Mathematically we calculate this by $\langle x \rangle = \langle x_2(t_2) - x_1(t_1) \rangle$. We found from our simulations that the larger the team of motors pulling the cargo the longer the average run length before the cargo detaches from the filament, as shown in figure 13a. Moreover, we plot histograms of the distributions of run length for different values of $N$ in figure 13b. For a few motors $N = 1$, 2 and 3, we see that the cargo often moves backwards in addition to the preferred forwards direction whereas the cargo with $N = 10$ motors in figure 13b almost always moves in the positive direction. For multiple motors, the run length appears to be roughly exponential in the number of motors, which corresponds with the conclusion of [4,14].

Recently Wilson *et al.* [66] have reported that cargo diffusion shortens the run length for single kinesin-1 leading to a non-monotonic relationship with increasing cargo viscous drag in simulations. This effect is due to load dependence of off rates such that at low viscous drag cargo diffusion can assist the motor direction whereas at high viscous drag the run length is shortened because of the viscous drag of the load. Assuming the viscosity in cytoplasm is more than 10 times that of water and

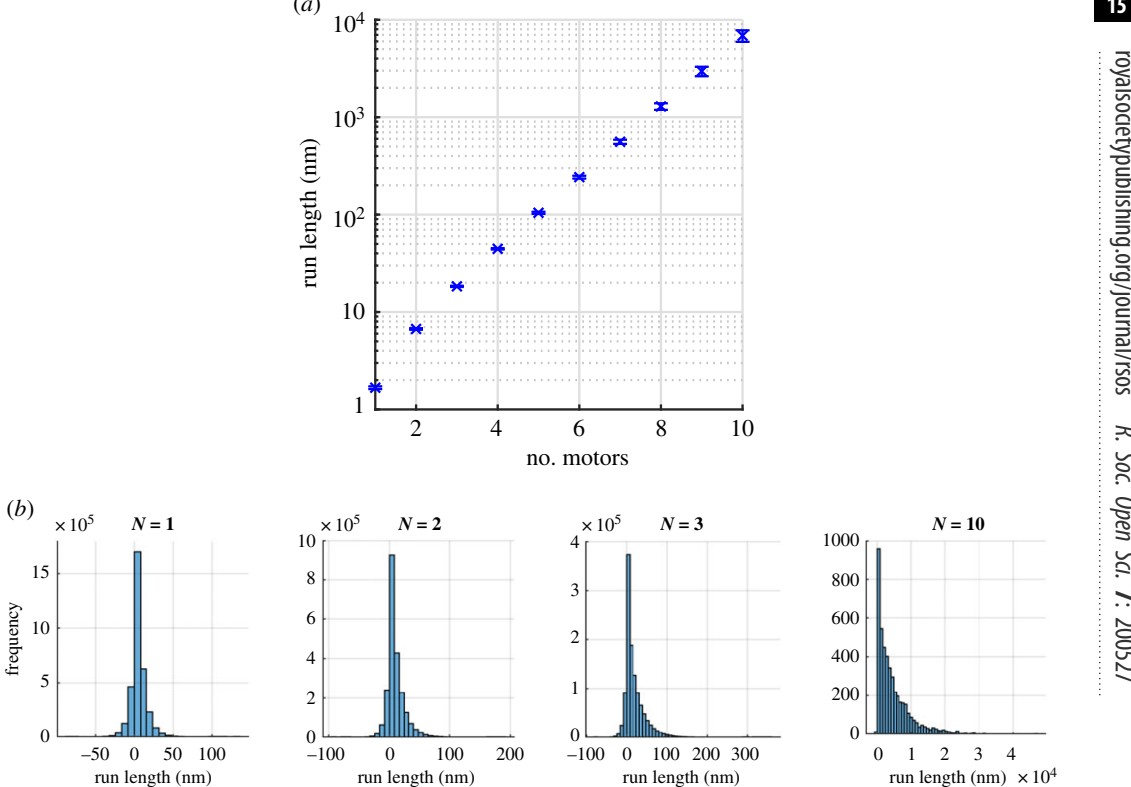

**Figure 13.** (a) Average run lengths of $N = 1$ to $N = 10$ non-processive motors from 100 Monte Carlo simulation runs with $f = 2$. The other parameter values are $k_{on} = 20$ s$^{-1}$, $k_{off} = 10$ s$^{-1}$, $p = 22$ s$^{-1}$, $q = 2.2$ s$^{-1}$ and $\delta = 0.5$. (b) Histograms of the distributions of run lengths in (a) for $N = 1, 2, 3$ and $N = 10$. The total number of data is about 33 000, 22 000, 11 000 and 100 per simulation for $N = 1, 2, 3$ and $N = 10$, respectively.

the cargo size is more than 0.1 µm this low viscosity cargo diffusion effect is not important. Another suggestion of non-monotonic run length is given by Wang & Kolomeisky [67] who use first passage time calculations and Monte Carlo simulations to find the run length for two motors coupled by a spring. This spring coupling is a different scenario from what we study so not directly comparable.

# 4. Comparison with experimental results

In this section, we compare our theoretical and computational results with literature experimental data from two different systems.

## 4.1. In vitro experiment

First we compare our results with the *in vitro* experiments by Furuta *et al.* [10], who measured collective transport by processive kinesin-1 and non-processive Ncd motors. They constructed DNA–motor assemblies with a set number of motors varying from 1 to 4. They linked the motors with defined spacing by their DNA scaffold. They track the movement of the assemblies along microtubules by observing a fluorescent dye attached to the DNA scaffold using total internal reflection fluorescence microscopy. In their system there is no cargo, apart from the DNA linking the motors together, and we therefore assume the load force, $f$, is zero. To compare their experimental results with our model we ran simulations using parameters for kinesin-1 and Ncd as observed in the experiment (see below for details). We also compared the results with the analytical solution of equations (2.2) and (3.7) for processive and non-processive motors, respectively.

For the processive motor kinesin-1, the step length is 8 nm and the forward minus backwards stepping rate is found from the measured velocity to be $p - q = 98 \pm 0.6$ s$^{-1}$ for one motor and $p - q = 89 \pm 0.8$ s$^{-1}$ for multiple motors [10]. For no load, $f = 0$, the velocity given by equation (2.2) is independent from the motor number $N$. However, the experimental results of the velocity for kinesin-1

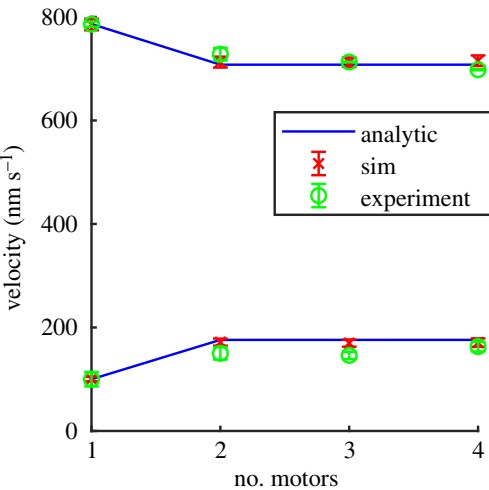

**Figure 14.** Velocity against number of motors to compare experimental data [10], simulation and analytical solution (equation (2.2)) for processive motor kinesin-1 (top) and equation (3.7) for non-processive motor Ncd (bottom), both with zero load, $f = 0$. For processive motors (top), parameters are $p = 110$ s$^{-1}$ for single kinesin-1 and $p = 100$ s$^{-1}$ for multiple kinesin-1. For non-processive motors (bottom), parameters are $p = 11$ s$^{-1}$ for single Ncd and $p = 22$ s$^{-1}$, $k_{on} = 20$ s$^{-1}$ and $k_{off} = 10$ s$^{-1}$ for multiple Ncds. $p = 10q$ for all cases.

in [10] show a slight decrease in velocity with increased number of motors. Furuta *et al.* [10] suggest that the motors in their assemblies slightly interfere with each other. The motors in their assemblies are coupled with DNA fragments so the motors are likely to experience attractive interactions at separations in which the elasticity of the linking DNA becomes important. This unknown interaction effect is not captured by our simple model. We therefore use different stepping rates for $N = 1$ and $N > 1$. Assuming $p = 10q$ [9], we use $p = 110$ s$^{-1}$ for $N = 1$ and $p = 100$ s$^{-1}$ for $N > 1$ in the plot in figure 14.

For the non-processive motor Ncd, the step length is also 8 nm and the forward stepping rate is $12.6 \pm 2.0$ s$^{-1}$ or $18.75 \pm 1.5$ s$^{-1}$ from the measured value for the velocity of one motor or two coupled. The results for the velocity from simulations and the analytical solution are plotted in figure 14 along with experimental data from [10] for both processive kinesin-1 and non-processive Ncd. Note that Furuta *et al.* [10] suggest that single Ncd has a lower forward stepping rate than when it is linked to other Ncd motors due to the unbound motors being kept close to the microtubule in the latter case. Our simulated and analytical results using the leading motor model in equation (3.7) match (within error bars) with their experimental data.

## 4.2. *In vivo* experiment

A significant practical problem of any motor transport experiment *in vivo* is to identify the numbers and types of motor proteins on the cargo. Therefore, comparing theoretical results with *in vivo* experimental data is more challenging because of the difficulty in determining the number of motors on the cargo, particularly in the case of non-processive motors. Several studies have indicated that many cellular cargoes *in vivo* are moved by a moderate number of motors ranging between one to five and the force exerted by the kinesin and dynein motors are mostly reported as about 1 pN [68–70].

To compare our results with an *in vivo* experiment, we used data from Pilling *et al.* [37], who tracked mitochondria in the axons of live *Drosophila* neurons. They tracked the movement of GFP-labelled mitochondria using scanning confocal fluorescence microscopy. They report that the mean $\pm$ s.d. of the net velocity of anterograde mitochondria (assumed to be moved by kinesin-1) was $0.26 \pm 0.10$ µms$^{-1}$. They found the mean $\pm$ s.d. run length for anterograde mitochondria was $1.82 \pm 1.19$ µm. It is difficult to know *in vivo* how many motors are attached to a cargo such as a mitochodrion. However, using our model, we can predict the expected number of kinesin-1 motors carrying the axonal mitrochondria in Pilling *et al.*'s [37] experiment. We use the same stepping rate parameters for kinesin-1 that we use in earlier parts of this paper, i.e. $p = 100$ s$^{-1}$ and $q = 10$ s$^{-1}$. However, as is seen in trajectories in [37], axonal kinesin-1 is slightly non-processive so we use $k_{on} = 5$ s$^{-1}$ and $k_{off} = 1$ s$^{-1}$ as proposed in [4]. We used $f = 2$ in our simulation corresponding to $F = fk_BT/dx = 1.1$ pN. We calculate the distribution of velocity and run length from our model for different number of motors. Figure 15*a*

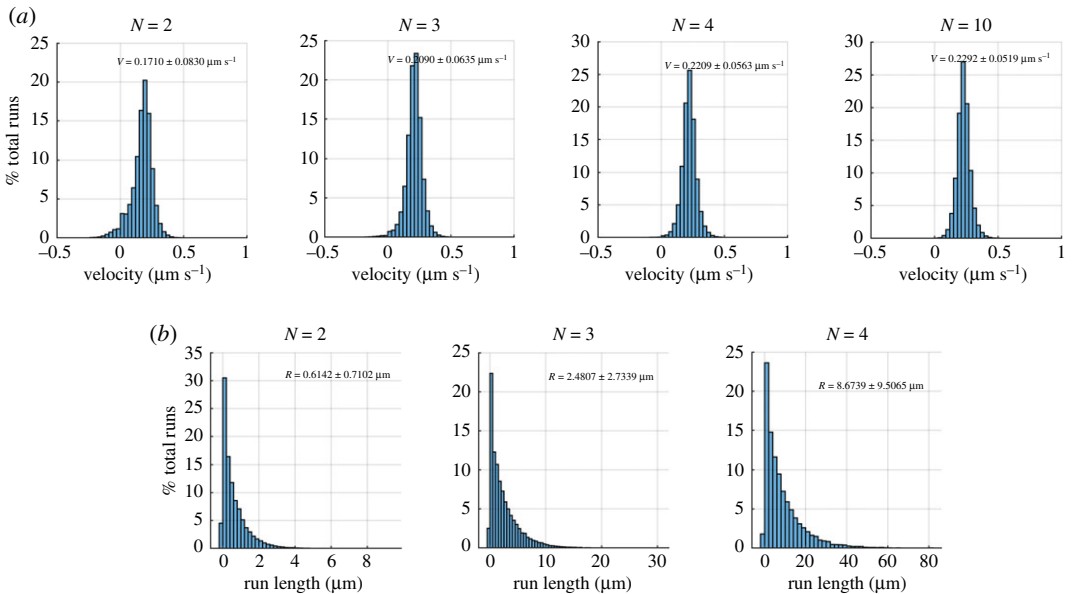

**Figure 15.** From our simulation, (*a*) velocity distribution for $N = 2$, 3, 4 and 10 kinesin-1 and (*b*) run length distributions for $N = 2$, 3 and 4 kinesin-1. The parameters are $p = 100$ s$^{-1}$, $q = 10$ s$^{-1}$, $k_{on} = 5$ s$^{-1}$ and $k_{off} = 1$ s$^{-1}$ [4].

shows that our simulations display almost always positive velocity when $N \geq 3$. The histograms plotted in figure 15 indicate that our model for $N = 3$ (velocity $= 0.21 \pm 0.06$ μm s$^{-1}$, run length $= 2.48 \pm 2.73$ μm) corresponds best with the experimental measurements from [37] (velocity $= 0.26 \pm 0.10$ μm s$^{-1}$, run length $= 1.82 \pm 1.19$ μm). Therefore, our simulation predicts that the expected number of motors carrying axonal mitochondria in [37] is three kinesin-1.

# 5. Conclusion

In this work, we have shown by analytical modelling and stochastic simulations that the velocity of cargo pulled by a cluster of molecular motors is insensitive to the number of motors $N$ bound to the cargo for $N > 1$ for processive motors and $N > 5$ for non-processive motors. Therefore, a cell investing in more than a handful of molecular motors on a cargo will not result in faster cargo transport. However, we find that the stall force is linear with the number of motors in the cluster for processive motors and also for large ($N > 10$) clusters of non-processive motors. Therefore, if a cargo is subjected to large forces, investing in more than a handful of molecular motors will enable a cell to overcome larger forces.

As expected, we find that the average velocity of cargoes pulled by small ($N < 5$) clusters of non-processive motors is significantly less than the velocity of the same number of processive motors. However, significantly, we show that for large number of motors, $N > 5$, clusters of non-processive motors approach the same velocity as those of clusters of processive motors (figure 11). For cargoes pulled by large, $N > 5$, clusters of non-processive motors the average velocity is insensitive to the detachment of motors since there is almost always at least one motor bound and therefore large non-processive motor clusters have the same velocity as for processive motors (figure 10). Therefore, according to our model and parameters used, a cluster of $N = 5$ non-processive motors is as fast at pulling a cargo as two processive motors. This is a clear indication of the effectiveness of non-processive motors for cellular transport.

A natural extension to our work would be to include force dependence in the unbinding rate $k_{off}$ as has been done in [1,14,51] and recently by Ucar & Lipowsky [71]. We would expect that including this effect would increase the force sensitivity, increase the time taken to reach steady state, decrease the stall force and decrease the mean velocity due to the zipper effect of successive leading motors experiencing the force-dependent unbinding.

We have extended the analytical model to the novel case of non-processive motors with limited number of binding sites, since for motors bound to a cargo the number of binding sites they can access on a filament may be severely limited by the cargo size. Limiting the number of binding sites $M$ decreases the number of non-processive motors bound to the filament. Our analytical expression

for the probability distribution of number of bound motors with limited binding sites correctly reproduces that for unlimited binding sites for $M \gg N$. We find that if the number of binding sites are limited, despite the clear shift in the probability distribution to smaller number of bound motors, the velocity is insensitive to this difference in probability distributions for small $N$ (figure 9). For large $N$, the velocity of a cluster of non-processive motors with limited binding sites approaches that of a cluster of processive motors. Our simulation results of the distribution of number bound motors fit better with the analytical distribution for limited number of binding sites than that for unlimited binding sites. However, since for small clusters the effect of limiting number of binding sites makes only a small difference to the average velocity, the distribution with unlimited number of binding sites is sufficient for calculating the average velocity of small clusters of non-processive motors and it has the advantage of one fewer parameters.

Our run length studies show that for small $N$ the cargo can move backwards as well as forwards but for $N \geq 10$ it almost always moves forwards. These findings may be of significant importance in interpreting *in vivo* experiments when assumptions are made about the identity of molecular motor types based on the observed direction of cargo. Enough data should be obtained to make a judgement based on statistically significant values given the stochasticity of the system.

Our simulations show the effect of sequence preservation which shifts the probability of bound motors to lower number of motors and decreases the stall force. This effect is larger for larger forces and larger clusters.

Finally, our model fits well with *in vitro* experiments by Furuta *et al.* [10] and predicts $N = 3$ as the most likely number of kinesin-1 motors bound to axonal mitochondria *in vivo* in the experiments by Pilling *et al.* [37].

Data accessibility. Code is available at the GitLab public repository, Project ID: 17661860: https://gitlab.com/Naruemon2532/processive-and-nonprocessive-motors.git (doi:10.5281/zenodo.3960327).

Competing interests. We declare we have no competing interests.

Funding. This work was supported by the Development and Promotion of Science and Technology Talents Project (Royal Government of Thailand scholarship) for NR, EPSRC E76039J Standard Research Studentship DTG for IDE, EPSRC grant no. EP/L026848/1 and the University of Sheffield for R.J.H.

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
