## [Reviewer comments · Royal Society Open Science]

Review History

RSOS-200527.R0 (Original submission)

Review form: Reviewer 1

Is the manuscript scientifically sound in its present form?

Yes

Are the interpretations and conclusions justified by the results?

Yes

Is the language acceptable?

Yes

Do you have any ethical concerns with this paper?

No

Have you any concerns about statistical analyses in this paper?

No

Recommendation?

Accept with minor revision (please list in comments)

Comments to the Author(s)

The manuscript of Rueangkham and coworkers presents a study of cargo transport in a cell by clusters of non-processive motors. The study includes analytical theory, simulations and a detailed comparison with experiments. The theory combines a statistical analysis of the attachment of the motors with two different scenarios to an existing asymmetric size exclusion model of the motors. The paper is well presented and the choice of the model well argued. I believe that this manuscript is an interesting contribution to the study of intracellular transport by molecular motors and I recommend publication in Royal Society Open Science after the authors take into account the following comments.

-Klumpp and Lipowsky also did thorough comparisons between theory and experiments for cargo transport by molecular motors. How does the current model compare to their approach, and how better is it to explain the experiments?

-The attachment and detachment of the motors treats the motors as completely independent and ignores any correlations between motors. There could for example exist correlations between the unbinding of one motor and the unbinding of the neighboring motor or a different probability of binding at a site where a motor is already bound. Is it clear that such correlations are negligible and what would they change?

-As the stepping rates, the binding and unbinding rates could depend on the load of the motors (and in general do). How would this affect the results?

Review form: Reviewer 2

Is the manuscript scientifically sound in its present form?

Yes

Are the interpretations and conclusions justified by the results?

Yes

Is the language acceptable?

Yes

Do you have any ethical concerns with this paper?

No

Have you any concerns about statistical analyses in this paper?

No

Recommendation?

Accept with minor revision (please list in comments)

Comments to the Author(s)

Review for: Modelling cytoskeletal transport by clusters of non-processive molecular motors with limited binding sites

The paper explores a leading motor model for processive and non-processive molecular motors. It does so through the comparison of analytic calculations with simulations. The novel result is the extension of the analytic expressions for the non-processive motors to the case of finite binding sites, which is then validated by simulations of the corresponding system. Furthermore, the paper discusses the effects of motor sequence preservation by implementing it in the simulations and

showing deviations from the analytic model without sequence preservation. The general conclusion is that the use of analytic models should be satisfactory for describing the number of bound motors and force-velocity curves of the systems investigated in the paper.

The quality of the presentation of the paper is good. The material is interesting and the writing is clear and sound. I recommend the publication of the manuscript after the following points are taken into account:

\subsection{Major points}

1) Eq. (3) is used for calculating the velocity of non-processive motors as an expected value of the steady-state velocities of processive motors. There is an implicit assumption made in this equation that a constant number of bound non-processive motors n , attain a well-defined steady-state velocity (that corresponds to the steady-state velocity of n processive motors). The validity of this assumption should depend on the ratio of the hopping and (un)binding time-scales. Can the authors show the validity of this assumption in the parameter regime used? In which parameter regime does this steady-state assumption break? How do the results compare to the force-velocity curves for non-processive motors without sequence preservation?

2) In the case of non-processing motors with variable binding sites and sequence preservation, how does the accuracy of the approximation of simulated P_n by the analytic eq. 8 (with average M) depend on the total number of motors N ? Can it be said that the accuracy monotonously increases with N , or it shows some non-monotonous trends? Could you draw a measure of accuracy versus N ?

In Figs. 9a2-9a3 it seems that the accuracy increases with N (from $N=3$ to $N=5$). However, in Fig. 8 for $N=10$ it seems that the accuracy is worse than for $N=5$. Could you comment on that in the light of the overall accuracy trends?

3) It seems that the forces used in Fig. 7 are not specified. Can this be rectified in the figure caption?

4) Can the number of binding sites M in the B scenario for non-processive motors, increase indefinitely by including the front/back binding sites? Due to the finite length of motors, this should become nonphysical at some point. When is this nonphysical regime reached? What would be the effect of introducing an upper cutoff for M ?

5) The force data for $N > 10$ should be provided since they are referenced in the conclusion.

6) Is it possible to develop a more quantitative method for deciding the best fit for the in vivo experiment? What modifications to the model could be envisaged to improve the match to the experiments?

7) In the leading motor model with processive motors, non-leading motors are not in any way affecting the leading motor hopping rates, or the load force applied on the leading motor. However, they are affecting the leading motor velocity through the exclusion assumption, that is by blocking the leading motor from backward jumping when occupying the site behind it. Would the force-velocity curve for the case without exclusion be independent of the motor number (for $N > 1$)? Could the force-velocity curve of the system with exclusion be reproduced by a single motor system with a modified backward rate? If so, could this line of thinking be extended to non-processive motors?

8) In the paper "Collective Dynamics of Interacting Molecular Motors", which derives the analytic expression (eq. 2), it is stated that this expression is derived in the "neutral" case. However, there are two more cases studied through simulations in that paper, the so-called "attractive" and "repulsive". Have there been any attempts to derive the analytic expressions for those two cases,

both for processive and non-processive motors, and explore the effect of finite binding sites in those cases?

9) In the simulation of non-processive motors, motors are allowed to bind with equal probability, linearly dependent on the number of unoccupied sites, at any unoccupied filament site. This seems like a reasonable assumption when binders are abundant. However if binders are scarce and spatially confined (motors bound to the cargo), this assumption may break, since it follows that an unoccupied site far from the motor increases its probability to bind to the closest site. In a more realistic scenario, the probability of binding is related to the distance between the motor and the binding site. Is it possible to discuss how would a non-constant probability distribution over unoccupied sites, that reflects spatial information, affect the results?

10) Can you give a short comment on the expected effect of the following modifications of the system:

Force-dependent (un)binding rates?

Finite motor length, which affects possible binding sites and binding rates?

Going beyond the leading motor model to force-dependent hopping rates of all motors, not just the leading one?

Decision letter (RSOS-200527.R0)

Dear Dr Hawkins,

On behalf of the Editors, I am pleased to inform you that your Manuscript RSOS-200527 entitled "Modelling cytoskeletal transport by clusters of non-processive molecular motors with limited binding sites" has been accepted for publication in Royal Society Open Science subject to minor revision in accordance with the referee suggestions. Please find the referees' comments at the end of this email.

The reviewers and handling editors have recommended publication, but also suggest some minor revisions to your manuscript. Therefore, I invite you to respond to the comments and revise your manuscript.

- Ethics statement

- Data accessibility

It is a condition of publication that all supporting data are made available either as supplementary information or preferably in a suitable permanent repository. The data accessibility section should state where the article's supporting data can be accessed. This section should also include details, where possible of where to access other relevant research materials such as statistical tools, protocols, software etc can be accessed. If the data has been deposited in an external repository this section should list the database, accession number and link to the DOI for all data from the article that has been made publicly available. Data sets that have been

deposited in an external repository and have a DOI should also be appropriately cited in the manuscript and included in the reference list.

If you wish to submit your supporting data or code to Dryad (<http://datadryad.org/>), or modify your current submission to dryad, please use the following link:
<http://datadryad.org/submit?journalID=RSOS&manu=RSOS-200527>

- **Competing interests**

- **Authors' contributions**

- **Acknowledgements**

- **Funding statement**

Because the schedule for publication is very tight, it is a condition of publication that you submit the revised version of your manuscript before 07-Jun-2020. Please note that the revision deadline will expire at 00.00am on this date. If you do not think you will be able to meet this date please let me know immediately.

When submitting your revised manuscript, you will be able to respond to the comments made by the referees and upload a file "Response to Referees" in "Section 6 - File Upload". You can use this to document any changes you make to the original manuscript. In order to expedite the

processing of the revised manuscript, please be as specific as possible in your response to the referees. We strongly recommend uploading two versions of your revised manuscript:

If your manuscript is newly submitted and subsequently accepted for publication, you will be asked to pay the article processing charge, unless you request a waiver and this is approved by Royal Society Publishing. You can find out more about the charges at <https://royalsocietypublishing.org/rsos/charges>. Should you have any queries, please contact openscience@royalsociety.org.

Kind regards,
Lianne Parkhouse
Editorial Coordinator
Royal Society Open Science
openscience@royalsociety.org

on behalf of Dr Andrew Angel (Associate Editor) and Pietro Cicuta (Subject Editor)

Associate Editor Comments to Author (Dr Andrew Angel):

I am recommending that the manuscript be accepted following satisfactory minor revisions in response to the reviewers suggestions.

Reviewer comments to Author:

Reviewer: 1

Comments to the Author(s)

The manuscript of Rueangkham and coworkers presents a study of cargo transport in a cell by clusters of non-processive motors. The study includes analytical theory, simulations and a detailed comparison with experiments. The theory combines a statistical analysis of the attachment of the motors with two different scenarios to an existing asymmetric size exclusion model of the motors. The paper is well presented and the choice of the model well argued. I believe that this manuscript is an interesting contribution to the study of intracellular transport by molecular motors and I recommend publication in Royal Society Open Science after the authors take into account the following comments.

-Klumpp and Lipowsky also did thorough comparisons between theory and experiments for cargo transport by molecular motors. How does the current model compare to their approach, and how better is it to explain the experiments?

-The attachment and detachment of the motors treats the motors as completely independent and ignores any correlations between motors. There could for example exist correlations between the unbinding of one motor and the unbinding of the neighboring motor or a different probability of binding at a site where a motor is already bound. Is it clear that such correlations are negligible and what would they change?

-As the stepping rates, the binding and unbinding rates could depend on the load of the motors (and in general do). How would this affect the results?

Reviewer: 2

Comments to the Author(s)

Review for: Modelling cytoskeletal transport by clusters of non-processive molecular motors with limited binding sites

The paper explores a leading motor model for processive and non-processive molecular motors. It does so through the comparison of analytic calculations with simulations. The novel result is the extension of the analytic expressions for the non-processive motors to the case of finite binding sites, which is then validated by simulations of the corresponding system. Furthermore, the paper discusses the effects of motor sequence preservation by implementing it in the simulations and showing deviations from the analytic model without sequence preservation. The general conclusion is that the use of analytic models should be satisfactory for describing the number of bound motors and force-velocity curves of the systems investigated in the paper.

The quality of the presentation of the paper is good. The material is interesting and the writing is clear and sound. I recommend the publication of the manuscript after the following points are taken into account:

\subsection{Major points}

1) Eq. (3) is used for calculating the velocity of non-processive motors as an expected value of the steady-state velocities of processive motors. There is an implicit assumption made in this equation that a constant number of bound non-processive motors n , attain a well-defined steady-state velocity (that corresponds to the steady-state velocity of n processive motors). The validity of this assumption should depend on the ratio of the hopping and (un)binding time-scales. Can the authors show the validity of this assumption in the parameter regime used? In which parameter regime does this steady-state assumption break? How do the results compare to the force-velocity curves for non-processive motors without sequence preservation?

2) In the case of non-processing motors with variable binding sites and sequence preservation, how does the accuracy of the approximation of simulated $P_{\{n\}}$ by the analytic eq. 8 (with average M) depend on the total number of motors N ? Can it be said that the accuracy monotonously increases with N , or it shows some non-monotonous trends? Could you draw a measure of accuracy versus N ?

In Figs. 9a2-9a3 it seems that the accuracy increases with N (from $N=3$ to $N=5$). However, in Fig. 8 for $N=10$ it seems that the accuracy is worse than for $N=5$. Could you comment on that in the light of the overall accuracy trends?

3) It seems that the forces used in Fig. 7 are not specified. Can this be rectified in the figure caption?

4) Can the number of binding sites M in the B scenario for non-processive motors, increase indefinitely by including the front/back binding sites? Due to the finite length of motors, this should become nonphysical at some point. When is this nonphysical regime reached? What would be the effect of introducing an upper cutoff for M ?

5) The force data for $N > 10$ should be provided since they are referenced in the conclusion.

6) Is it possible to develop a more quantitative method for deciding the best fit for the in vivo experiment? What modifications to the model could be envisaged to improve the match to the experiments?

7) In the leading motor model with processive motors, non-leading motors are not in any way affecting the leading motor hopping rates, or the load force applied on the leading motor. However, they are affecting the leading motor velocity through the exclusion assumption, that is by blocking the leading motor from backward jumping when occupying the site behind it. Would the force-velocity curve for the case without exclusion be independent of the motor number (for $N > 1$)? Could the force-velocity curve of the system with exclusion be reproduced by a single motor system with a modified backward rate? If so, could this line of thinking be extended to non-processive motors?

8) In the paper "Collective Dynamics of Interacting Molecular Motors", which derives the analytic expression (eq. 2), it is stated that this expression is derived in the "neutral" case. However, there are two more cases studied through simulations in that paper, the so-called "attractive" and "repulsive". Have there been any attempts to derive the analytic expressions for those two cases, both for processive and non-processive motors, and explore the effect of finite binding sites in those cases?

9) In the simulation of non-processive motors, motors are allowed to bind with equal probability, linearly dependent on the number of unoccupied sites, at any unoccupied filament site. This seems like a reasonable assumption when binders are abundant. However if binders are scarce and spatially confined (motors bound to the cargo), this assumption may break, since it follows that an unoccupied site far from the motor increases its probability to bind to the closest site. In a more realistic scenario, the probability of binding is related to the distance between the motor

and the binding site. Is it possible to discuss how would a non-constant probability distribution over unoccupied sites, that reflects spatial information, affect the results?

10) Can you give a short comment on the expected effect of the following modifications of the system:

Force-dependent (un)binding rates?

Finite motor length, which affects possible binding sites and binding rates?

Going beyond the leading motor model to force-dependent hopping rates of all motors, not just the leading one?

Author's Response to Decision Letter for (RSOS-200527.R0)

See Appendix A.

Decision letter (RSOS-200527.R1)

Dear Dr Hawkins,

It is a pleasure to accept your manuscript entitled "Modelling cytoskeletal transport by clusters of non-processive molecular motors with limited binding sites" in its current form for publication in Royal Society Open Science. The comments of the reviewer(s) who reviewed your manuscript are included at the foot of this letter.

Kind regards,

Anita Kristiansen
Editorial Coordinator

on behalf of Dr Andrew Angel (Associate Editor) and Pietro Cicuta (Subject Editor)
openscience@royalsociety.org

Associate Editor Comments to Author (Dr Andrew Angel):

Comments to the Author:

As far as I can see, solid efforts have been made to address all of the referees' suggested revisions. I am therefore recommending this manuscript for acceptance as is.

Appendix A

Dr Rhoda Hawkins
Department of Physics and Astronomy
University of Sheffield
Hicks Building, Hounsfield Road
Sheffield, S3 7RH, UK

June 23, 2020

Editor
Royal Society Open Science

Dear Editor,

We thank you for accepting our manuscript subject to minor revision in accordance with the referee suggestions. Here we provide a detailed response to those referee suggestions (copied here in italics for convenience).

Response to referee 1

-Klumpp and Lipowsky also did thorough comparisons between theory and experiments for cargo transport by molecular motors. How does the current model compares to their approach, and how better is it to explain the experiments?

Klumpp et al PNAS 2005 used the same model for the distribution of bound motors as us but they used a mean field model in which the force is equally shared between all motors whereas we use the more general leading motor model which we think is more appropriate in vivo. As we mention in our manuscript their linear mean field model is equivalent to our exponential force dependence in the limit of small forces. The main findings are not very sensitive to this distinction, especially for small forces. Whilst in their PNAS 2005 Klumpp et al presented analytical calculations, in their PNAS 2008 Muller et al. they also did Gillespie simulations. Our model includes two major extensions to that of Klumpp and Lipowsky. We include limited binding sites in both analytical calculations and simulations and sequence preservation in simulations. We have modified the end of section I to make this comparison with the work of Klumpp and Lipowsky more explicit.

-The attachment and detachment of the motors treats the motors as completely independent and ignores any correlations between motors. There could for example exist correlations between the unbinding of one motor and the unbinding of the neighboring motor or a different probability of binding at a site where a motor is already bound. Is it clear that such correlations are negligible and what would they change?

In the simple exclusion process we implement the probability of binding to a site already bound by a motor is zero - we only allow binding to unoccupied sites. We did not include coupling between neighbouring motors in (un)binding to be consistent with our choice to consider the case of neutral

interactions between motors on the stepping rates. For stepping rates Campas et al (PRL 2006) include attractive or repulsive interactions and find differences in the velocities of motor clusters between these cases (slower/faster for attractive/repulsive). Despite a small amount of evidence suggesting very weak attractive interactions (Roos et al 2008), it is not yet clear what interactions are present experimentally. We therefore focus on the neutral case (no interactions) for our analysis. If positive cooperativity was introduced in motor (un)binding then we would expect tighter clusters of motors to form as is the case for attractive interactions in stepping. We expect cooperative unbinding to increase the stochastic effects (i.e. larger fluctuations) but not to change the mean steady state behaviour. We expect that increasing the probability of binding next to an existing motor would decrease the time taken to reach steady state but would otherwise not affect the results.

We have added a couple of sentences to the main text in section II.A.2. to address this point.

-As the stepping rates, the binding and unbinding rates could depend on the load of the motors (and in general do). How would this affect the results?

Yes the unbinding is likely to be force dependent. We would expect that including this effect would increase the force sensitivity, increase the time taken to reach steady state, decrease the stall force and decrease the mean velocity due to the zipper effect of successive leading motors force dependent unbinding. This effect has been studied by Campas et al 2008, Kunwar and Mogilner 2010 and recently by Ucar and Lipowsky 2020.

We have added a comment on this to section V.

Response to referee 2

- 1. Eq. (3) is used for calculating the velocity of non-processive motors as an expected value of the steady-state velocities of processive motors. There is an implicit assumption made in this equation that a constant number of bound non-processive motors n , attain a well-defined steady-state velocity (that corresponds to the steady-state velocity of n processive motors). The validity of this assumption should depend on the ratio of the hopping and (un)binding time-scales. Can the authors show the validity of this assumption in the parameter regime used? In which parameter regime does this steady-state assumption break? How do the results compare to the force-velocity curves for non-processive motors without sequence preservation?*

The referee is correct that Eq. (3) relies on this steady state assumption. That is why we wait for our simulations to reach steady state before measuring the velocity. As stated in the manuscript we take the steady state time $t_s = 2500$ steps for $N \leq 10$ non-processive motors. With the parameters we use the ratio of (un)binding to stepping rates is < 1 as required but it is not $\ll 1$. However the simple exclusion process (SEP) with sequence preservation that we include means that the binding rate is effectively much smaller and therefore our effective ratio is $\ll 1$. If this assumption is broken then the analytical solution gives an over estimate since the steady state SEP cluster has reduced backwards stepping

rates leading to faster forwards velocities compared to non-steady-state. This can be seen in Fig 9 for larger forces where the effective stepping rate is small due to the large forces. We have added a sentence following Eq 3 to make this assumption more explicitly and a comment on Figure 9.

2. *In the case of non-processing motors with variable binding sites and sequence preservation, how does the accuracy of the approximation of simulated P_n by the analytic eq. 8 (with average M) depend on the total number of motors N ? Can it be said that the accuracy monotonously increases with N , or it shows some non-monotonous trends? Could you draw a measure of accuracy versus N ?*

In Figs. 9a2-9a3 it seems that the accuracy increases with N (from $N=3$ to $N=5$). However, in Fig. 8 for $N=10$ it seems that the accuracy is worse than for $N=5$. Could you comment on that in the light of the overall accuracy trends?

The mismatch between the approximate $P(n)$ given by Eq 8 and the simulations does not directly depend on the total number of motors, N , but it does depends on how the cluster is bound on the track. Eq 8 with average M overestimates the number of motors bound compared to the simulations with binding restricted by sequence preservation. This increases with N since a larger proportion of the binding region M is affected by sequence preservation restrictions. As requested by the referee we provide here in Figure 1 a plot of a measure of the accuracy against N .

Figure 1. Difference in the probability distribution of bound motors against the total number of non-processive motors N for zero load force, $f = 0$. The difference, $P(n)_{sim} - P(n)_{analytic}$, is defined as the difference between the mean value of $P(n)$ from 100 simulation runs and the analytically calculated value $P(n)$ from Eq.8 using the mean value of M taken from the corresponding simulations. In the figure, each circle represents the accuracy of the point on the probability distribution with n bound motors out of the total N motors.

The plot shows the general trend of increasing mismatch with increasing N . However two of the $N = 3$ points have a larger magnitude mismatch than this trend would expect. These points correspond to $n = 1$ and $n = 3$. Figure 9(a2) in the main text shows that the analytical curve with binding sites limited to $M = 7$ is actually shifted to smaller numbers of

motors bound compared to the simulations. This is due to the end effects of allowing binding one site in front and one site behind the first/last motor on the filament in the simulations giving more freedom than the analytical equivalent of $M = 7$ fixed sites. These end sites are a greater proportion of the total number of sites available for small number of motors (29% for $N = 3$) than for larger number of motors (17% for $N = 5$, 1% for $N = 100$). Hence the mismatch due to the effect is most prominent for the $N = 3$ case.

We have added a comment on this to the discussion on Figure 9 in the main text.

3. *It seems that the forces used in Fig. 7 are not specified. Can this be rectified in the figure caption?*

The force ($f = 0$) has now been put in the captions of Fig. 7 and 8.

4. *Can the number of binding sites M in the B scenario for non-processive motors, increase indefinitely by including the front/back binding sites? Due to the finite length of motors, this should become nonphysical at some point. When is this nonphysical regime reached? What would be the effect of introducing an upper cutoff for M ?*

The number of binding sites M is defined as the distance between the first and last bound motors plus 2 (the latter for the front/back binding sites). This does not go to infinity because the simple exclusion process (SEP) causes the motors to form a cluster at steady state. The maximum value for M is therefore the steady state cluster size plus 2. The minimum value for M is the number of bound motors plus 2 i.e. $n + 2$. The nonphysical regime considered by the referee would be reached if repulsive interactions between motors were included (we don't consider such interactions). The effect of introducing an upper cutoff for M is more easily seen in our scenario A where various fixed values for M are compared to unlimited binding sites in Figure 7(b).

We have added a sentence to clarify this point in section III.C.1.b.

5. *The force data for $N > 10$ should be provided since they are referenced in the conclusion.*

Figure 2. Extended version of manuscript Figure 12 including data for $10 \leq N \leq 100$ on a log-log plot.

The additional data shown here is evidence for the claim we made in our conclusion “the stall force is linear with the number of motors in the cluster for processive motors and also for large ($N > 10$) clusters of non-processive motors”. However since we have only checked the expected linear trend for $N > 10$ and not systematically run simulations for every point we do not wish to include this extra data in the main manuscript. We do not think this point is worth spending the extra computational time to run the simulations for all the points given that the number of motors expected in vivo is small (Gross et al 2007). We therefore provide this extra data in the appendix.

As stated in the manuscript text, the analytical expression for processive motors shows the stall force is linear with the number of motors in the cluster. In section III.C.2. of the manuscript we argue that for large ($N > 10$) clusters of non-processive motors the velocity is equivalent to that of the same number of processive motors. We therefore expect the linear relationship for the stall force for processive motors to also hold for non-processive motors. Our data shows that for the analytical calculations for non processive motors the large N trend is linear with the same gradient as the processive motor case. The analytical calculation including limited binding sites using the average value obtained from simulations shows only slightly lower stall force compared to the unlimited case. Interestingly however the simulations including sequence preservation show a much lower gradient indicating that the sequence preservation has a significant effect on the stall force due to the restrictions on binding caused by the sequence preservation.

We have added references to the data now included in the Appendix at appropriate points in the main text.

6. *Is it possible to develop a more quantitative method for deciding the best fit for the in vivo experiment? What modifications to the model could be envisaged to improve the match to the experiments?*

In response to this point from the referee we decided to remove the focus on qualitative comparison between histograms and instead quote the mean and standard deviations of the experimental and our theoretical results for a more quantitative comparison. The relevant figure and text has been adapted to do this.

7. *In the leading motor model with processive motors, non-leading motors are not in any way affecting the leading motor hopping rates, or the load force applied on the leading motor. However, they are affecting the leading motor velocity through the exclusion assumption, that is by blocking the leading motor from backward jumping when occupying the site behind it. Would the force-velocity curve for the case without exclusion be independent of the motor number (for $N > 1$)? Could the force-velocity curve of the system with exclusion be reproduced by a single motor system with a modified backward rate? If so, could this line of thinking be extended to non-processive motors?*

Yes the force-velocity curve for the case without exclusion would be independent of the motor number i.e. the same as that of a single motor.

We show here, and in the appendix of the manuscript, an example of force-velocity curves for the case $q \ll pe^{-f/2}$ for which we can approximate equation (2) by $V_N \approx pe^{-f/2} - qe^{f/2}(q/p)^{N-1}$ which is the single motor case with the backwards stepping rate modified by

the factor $(q/p)^{N-1}$. This can reproduce an approximation to the curves seen in figure 3 but has the disadvantage of only being valid for $q \ll pe^{-f/2}$ which is not well satisfied for our parameter values.

Figure 3. Force-velocity curves of $N = 1$ (blue), 2 (red) and 10 (green) processive motors using the full analytical expression from the main text Eq 2. (solid lines) and an approximate analytical expression $V_N \approx pe^{-f/2} - qe^{f/2}(q/p)^{N-1}$ which corresponds to a single motor with backwards stepping rate modified by the factor $(q/p)^{N-1}$ (dashed lines). The equivalent simulation points are shown on the main text Figure 3.

If one was to extend this to non-processive motors it would only be possible by assuming an average value for the number of bound motors n and therefore it would not be possible to study stochastic effects due to (un)binding with such a model.

8. In the paper "Collective Dynamics of Interacting Molecular Motors", which derives the analytic expression (eq. 2), it is stated that this expression is derived in the "neutral" case. However, there are two more cases studied through simulations in that paper, the so-called "attractive" and "repulsive". Have there been any attempts to derive the analytic expressions for those two cases, both for processive and non-processive motors, and explore the effect of finite binding sites in those cases?

Campas et al (PRL 2006) analytically calculate the velocity for 2 motors for attractive, repulsive and neutral interactions cases. However as far as we are aware this has not been done analytically for the general case of N processive nor non-processive motors. However Campas et al (PRL 2006) themselves studied these cases computationally. Despite a small amount of evidence suggesting very weak attractive interactions (Roos et al 2008), it is not yet clear what interactions are present experimentally. We therefore focus on the neutral case of no interactions between the motors for our analysis. We have re-added explicit mention of these interaction cases in section II.A.2.

9. In the simulation of non-processive motors, motors are allowed to bind with equal probability, linearly dependent on the number of unoccupied sites, at any unoccupied filament site. This

seems like a reasonable assumption when binders are abundant. However if binders are scarce and spatially confined (motors bound to the cargo), this assumption may break, since it follows that an unoccupied site far from the motor increases its probability to bind to the closest site. In a more realistic scenario, the probability of binding is related to the distance between the motor and the binding site. Is it possible to discuss how would a non-constant probability distribution over unoccupied sites, that reflects spatial information, affect the results?

It is precisely because motors are spatially restricted when bound to a cargo that we studied the case of limiting the number of binding sites to only those close to the cargo. By limiting the binding sites available to the motor we introduce a non-constant probability distribution of binding sites in space in the form of a top hat (rectangular) function with the binding rate k_{on}^s for each site within the binding region and zero binding probability outside of that region. The case of the binding probability varying linearly with distance is equivalent to a model in which the motors are treated as springs such as in Kunwar et al 2008, however in these studies it is the unbinding rate that is dependent on the spring force of stretching the motor with the binding rate assumed constant. We have added a comment when we first introduce limited binding sites to make this clearer.

10. *Can you give a short comment on the expected effect of the following modifications of the system:*

- *Force-dependent (un)binding rates?*

We would expect that including this effect would increase the force sensitivity, increase the time taken to reach steady state, decrease the stall force and decrease the mean velocity due to the zipper effect of successive leading motors force dependent unbinding unbinding. This effect has been studied by Campas et al 2008, Kunwar and Mogilner 2010 and recently by Ucar and Lipowsky 2020. We have now commented on this in section V.

- *Finite motor length, which affects possible binding sites and binding rates?*

We show in our work that limiting the available binding sites shifts the probability distribution of bound motors to smaller number of bound motors. However we show that the velocity of clusters of small ($N < 10$) total number of motors is insensitive to the different in probability distributions of bound motors. For large $N > 10$ clusters they rarely have all motors unbound and their velocity can be described by that of processive motors. For the case of treating a motor with a finite length modelled as a spring we refer the referee to Kunwar et al 2008 and Kunwar and Mogilner 2010. We have added a comment when we first introduce limited binding sites to make this clearer.

- *Going beyond the leading motor model to force-dependent hopping rates of all motors, not just the leading one?*

As we described in our manuscript (section II.A.1.) the mean field model (Klump and Lipowsky PNAS 2005) assumes all the motors have equal force dependent hopping rates since they share the load equally. We assume this is not what the referee means by “going beyond the leading motor model” and assume the referee means a model which is inbetween the two extremes of equal force sharing and the leading motor feels all the load. We would like to point the referee to the article by Kunwar et. al. 2008 in which they developed a Monte Carlo model of transport by multiple motors which share load unevenly. They model each motor as a spring exerting a restoring force when stretched beyond its natural length but no resistance to compression. From this they find that when motors are close together they share the load almost equally (as in the mean field model) but when the leading motor is ahead it experiences most of the force. In the limit of infinite spring stiffness the leading motor experiences all the force as in our model. Unlike us, in their model Kunwar et. al. 2008 do not include exclusion and allow more than one motor to be at the same position. When motors are at the same position they share the load equally as in the mean field model. Thus the Kunwar et. al. 2008 model combines features of both the leading motor and mean field models but does not include the simple exclusion process that we study. We have included a comment on this in section II.A.2.

Yours sincerely,

On behalf of all the authors:
Naruemon Rueangkham,
Ian Estabrook
and Rhoda Hawkins